



# Long-term reliability of the Figaro TGS 2600 solid-state methane sensor under low Arctic conditions at Toolik lake, Alaska

Werner Eugster[1], James Laundre[2], Jon Eugster[3,4], and George W. Kling[5]

[1]ETH Zurich, Department of Environmental Systems Science, Institute of Agricultural Sciences, Universitätstrasse 2, 8092 Zurich, Switzerland
[2]The Ecosystem Center, Marine Biology Laboratory, Woods Hole, MA 02543, USA
[3]University of Zurich, Institute of Mathematics, Winterthurerstrasse 190, 8057 Zurich, Switzerland
[4]now at: School of Mathematics, The University of Edinburgh
[5]University of Michigan, Department of Ecology & Evolutionary Biology, Ann Arbor, MI 48109-1085, USA

**Correspondence:** Werner Eugster (eugsterw@ethz.ch)

**Abstract.** The TGS 2600 was the first low-cost solid state sensor that shows a weak response to ambient levels of $CH_4$ (e.g., range ≈1.8–2.7 ppm). Here we present an empirical function to correct the TGS 2600 signal for temperature and (absolute) humidity effects and address the long-term reliability of two identical sensors deployed from 2012 to 2018. We assess the performance of the sensors at 30-minute resolution and aggregated to weekly medians. Over the entire period the agreement

between TGS-derived and reference $CH_4$ concentrations measured by a high-precision Los Gatos Research instrument was $R^2$ = 0.42, with better results during summer ($R^2$ = 0.65 in summer 2012). Using absolute instead of relative humidity for the correction of the TGS 2600 sensor signals reduced the typical deviation from the reference to less than ±0.1 ppm over the full range of temperatures from –41°C to 27°C. At weekly resolution the two sensors showed a downward drift of signal voltages indicating that after 10–13 years a TGS 2600 may have reached its end of life. While the true trend in $CH_4$ concentrations

measured by the high-quality reference instrument was 10.1 ppb $yr^{-1}$ (2012–2018), part of the downward trend in sensor signal (ca. 40–60%) may be due to the increase in $CH_4$ concentration, because the sensor voltage decreases with increasing $CH_4$ concentration. Weekly median diel cycles tend to agree surprisingly well between the TGS 2600 and reference measurements during the snow-free season, but in winter the agreement is lower. We suggest developing separate functions for deducing $CH_4$ concentrations from TGS 2600 measurements under cold and warm conditions. We conclude that the TGS 2600 sensor

can provide data of research-grade quality if it is adequately calibrated and placed in a suitable environment where cross-sensitivities to gases other than $CH_4$ is of no concern.

Keywords: Toolik lake, Northern Alaska, Methane, Trace gas sensor, SnO2 sensor, Leak detector.

## 1 Introduction

Low-cost trace gas sensors open new deployment opportunities for environmental observations. Still, their long-term perfor-

mance in real-world applications is largely unknown, and thus, scientific research with such low-cost sensors is challenged with a high risk of failure and questionable data quality. Hence low-cost sensors are only considered as a complementary





source of information on air quality (e.g. Lewis et al., 2018). Here we report on a six-year (2012–2018) deployment of two low-cost Figaro TGS 2600 methane ($CH_4$) sensors during summer and winter conditions in the relatively harsh low Arctic climate of northern Alaska to explore the long-term stability and reliability of $CH_4$ concentration estimates. The sensors were previously deployed over Toolik lake during the ice-free season 2011 (Eugster and Kling, 2012), where similar values between TGS-derived and reference $CH_4$ concentrations were only found if measurements were integrated over at least 6 hours, or if they were aggregated to mean diel cycles over the season. Other studies have deployed the same sensor type in complex rural and urban environments along the Colorado Front Range (Collier-Oxandale et al., 2018), in an oil and gas production region (Greeley, Colorado; Casey et al., 2019), and in urban South Los Angeles (Shamasunder et al., 2018). An application on an unmanned aerial vehicle, however, did not successfully detect $CH_4$ hotspots (Falabella et al., 2018). These are all pioneering studies, but mostly restricted to a few days to months of measurements. Thus, our study is the first long-term comparison of high-precision measurements to those from $CH_4$ sensitive, low-cost sensors under challenging climatic conditions.

Typically, new sensors are first calibrated under controlled conditions in a laboratory environment. Extensive calibration tests with a similar low-cost sensor (Figaro TGS2611-E00) from the same manufacturer as our TGS 2600 have been carried out by van den Bossche et al. (2017). Despite the care taken in their calibration effort, the residual $CH_4$ concentration after calibration was still on the order of $\pm 1.7$ ppm, which is acceptable for chamber flux measurements, e.g. over water (as e.g., done by Duc et al., 2019), but not sufficient to measure ambient atmospheric concentrations, which are of the same order of magnitude as the calibration uncertainty. The issue of important differences between laboratory assessments of low-cost sensor and their real-world performance is well known and typically relates to different data correction and calibration approaches in real-world than laboratory applications (Lewis et al., 2018). Hence we decided to use outdoor measurements obtained over a wide range of temperatures and relative humidity – the major cross-sensitivities experienced by such sensors – and derive a calibration function via parameter extraction using this data set. Our goals were thus to (1) establish a statistical calibration function from field measured conditions that can also be used in different contexts to linearize the TGS 2600 sensor signal (which then can still be fine-tuned with a two-point calibration in a specific application); (2) assess the reliability of the TGS 2600 low-cost sensor under winter and summer conditions in the Arctic over seven years of continuous deployment; and (3) explore potential improvements for sensor data processing, which includes (3a) wind effects that are neglected in laboratory environments, and (3b) artificial neural networks (ANN) to find out whether results can be improved over standard statistical regression methods for calibration of the sensor.

## 2 Material and methods

### 2.1 Study site

Field measurements were carried out at the Toolik wet sedge site (TWE, 68°37′27.62″ N, 149°36′08.10″ W, 728.14 m elevation, WGS 84 datum) where seasonal eddy covariance flux measurements were carried out during the summer seasons of 2010–2015 and partially during winters starting in 2014 until 15 June 2016, with the continuation as a meteorological station until present. The site is a wetland that is a local source of $CH_4$ with a flux rate that is roughly one order of magnitude





stronger than adjacent Toolik lake, where the Eugster and Kling (2012) study was performed. The site is a wet graminoid tundra dominated by sedge species, namely cotton grass (*Eriophorum angustifolium*) and *Carex aquatilis* (Walker and Everett, 1991).

## 2.2 Instrumentation and measurements

Two Figaro TGS 2600 sensors (Figaro, 2005b, a) that were already deployed over Toolik lake (TOL) during the ice-free season
2011 (Eugster and Kling, 2012) were installed at the TWE site in late June 2012 (Fig. 1). The TGS 2600 is a high-sensitivity solid-state sensor for the detection of air contaminants (Figaro, 2005a). It is sensitive to methane at low concentrations, but also to hydrogen, carbon monoxide, iso-butane, and ethanol. It is the only low-cost solid-state sensor that we are aware of for which the manufacturer indicates a sensitivity to methane even under ambient ($\approx$2 ppm) methane concentrations, whereas most other sensors are only sensitive at concentrations that exceed ambient levels by at least one or two orders of magnitude. This high
sensitivity to low methane concentrations comes at the expense that no specific molecular filter prevents the other components to reach the sensor surface. Thus, our considerations made here assume that deployment is made in an area like e.g., the Arctic where levels of carbon monoxide, iso-butane, ethanol and hydrogen are rather constant and do not vary as strongly as does methane, so that the sensor signal can be interpreted as a first approximation as a methane concentration signal. For additional details on the TGS 2600 sensor the reader is referred to Eugster and Kling (2012).

The TWE site receives line power from the Toolik Field Station (TFS) power generator. During the snow and ice-free summer season (typically late June to mid-August) measurements are almost interruption free, but during the cold season (typically September to late May) longer power interruptions limit the winter data coverage. Nevertheless, this is the first study that provides low-cost sensor methane concentration measurements over a temperature range from Arctic winter temperatures of –41°C and to a relatively balmy 27°C during short periods of the Arctic summer. Reference $CH_4$ concentrations were
measured by a Fast Methane Analyzer (FMA, Los Gatos Research, Inc., San Jose, CA, USA; years 2012–2016) which was replace by a Fast Greenhouse Gas Analyzer for combined $CH_4$/$CO_2$/$H_2O$ concentration measurements (FGGA, Los Gatos Research, Inc., San Jose, CA, U.S.A.; since 2016). Until 18 June 2016 the $CH_4$ concentrations were calculated as 1-minute averages from the raw eddy covariance flux data files. We report all gas concentrations in mixing ratios by volume (ppm or ppb dry mole fractions). The FMA and FGGA sampling rate was set to 20 Hz, and the flow rate of sample air was ca. 20 L min$^{-1}$.
After the termination of eddy covariance flux measurements, the FGGA measurements were continued with the instrument's internal pump (flow rate ca. 0.65 L min$^{-1}$) with 1 Hz raw data sampling. In addition to digital recording, the $CH_4$ signal was converted to an analog voltage that was recorded on a CR23X data logger (Campbell Scientific Inc. [CSI], Logan, UT, USA). The same data logger also recorded air temperature and relative humidity (HMP45AC, CSI), wind speed and wind direction (034B Windset, MetOne, Grants Pass, OR, USA), plus ancillary meteorological and soil variables not used in this
study. Sensors were measured every 5 seconds and 1-minute averages were stored on the logger. These data were then screened for outliers and instrumental errors and failures, and 30-minute averages were calculated for the present analysis.

Because the TGS 2600 sensors only show a weak response to $CH_4$, but are highly sensitive to temperature and humidity, a LinPicco A05 Basic sensor (IST Innovative Sensor Technology, Wattwil, Switzerland) was added next to the TGS 2600 (see





Fig. 1). The A05 is a capacitive humidity module that also has a Pt1000 platinum 1 kΩ thermistor on board to measure ambient

temperature. The relative humidity output by the A05 is a linearized voltage in the range 0–5 V, and the Pt1000 thermistor was

measured in three-wire half bridge mode using an excitation voltage of 4.897 V.

## 2.3   Calculations

The basic principle of operation of the TGS 2600 sensor was described in detail by Eugster and Kling (2012). The methane

sensing mechanisms of different active materials used in solid state sensors was described by Aghagoli and Ardyanian (2018).

The TGS 2600 uses a $SnO_2$ micro crystal surface (Figaro, 2005b). Whereas the manufacturer defines the sensor signal as

$R_s/R_0$, the ratio of the electrical resistance $R_s$ of the heated sensor material surface normalized over its resistance $R_0$ in the

air under absence of $CH_4$, Hu et al. (2016) define the sensor signal as the ratio between $R_s$ and $R_g$, the resistance of the surface

in pure gas of interest (here $CH_4$). In all cases, considerations of technical sensor information is made for high concentrations

of $CH_4$ (e.g., 200 ppm for a $SnO_2$ surface according to Hu et al. (2016), not for ambient concentrations in the typical range

1.7–4 ppm (or less). Hence, some adaptations are always necessary because present-day sensors are not yet designed for such

low concentrations. In order to simplify calculations compared to what we presented in Eugster and Kling (2012) – which

closely followed the technical information provided by the manufacturer (Figaro, 2005b, a) – we define the sensor signal as

$S_c = R_s/R_0$, but with $R_0$ arbitrarily set to the resistance observed when the sensor delivers $V_0 = 0.8$ V output at $V_c = 5.0$

V supply voltage. The highest voltages measured at TWE were 0.7501 and 0.7683 V from sensors #1 and #2, respectively

(which theoretically corresponds to the lowest $CH_4$ concentrations). With these assumptions the sensor signal $S_c$ can easily be

approximated as a function of the inverse of the measured TGS signal voltage $V_s$,

$$S_c = \frac{R_S}{R_0} \approx 0.952381 \cdot \frac{1}{V_s} - 0.1904762 \,. \tag{1}$$

The full derivation is

$$
\begin{aligned}
\frac{R_s}{R_0} &= \frac{\dfrac{V_c \cdot R_L}{V_s} - R_L}{\dfrac{V_c \cdot R_L}{V_0} - R_L} = \frac{\dfrac{V_c \cdot R_L - V_s \cdot R_L}{V_s}}{\dfrac{V_c \cdot R_L - V_s \cdot R_L}{V_0}} = \\
&= \frac{V_0 \left(V_c - V_s\right) R_L}{V_s \left(V_c - V_0\right) R_L} = \frac{V_0 \left(V_c - V_s\right)}{V_s \left(V_c - V_0\right)} = \\
&= \frac{V_0}{V_c - V_0}\left(\frac{V_c}{V_s} - 1\right) = \frac{1}{V_s} \cdot \frac{V_c \cdot V_0}{V_c - V_0} - \frac{V_0}{V_c - V_0} \,.
\end{aligned}
$$

Here $R_L$ is the load resistor over which $V_s$ is measured (see Figaro (2005a) or Eugster and Kling (2012), for more details),

but which can be eliminated in this algebraic simplification.

To compute absolute humidity, we used the Magnus equation to estimate saturation vapor pressure $e_{\text{sat}}$ (in hPa) at ambient

temperature $T_a$ (in °C),

$$e_{\text{sat}} = 6.107 \cdot 10^{a \cdot T_a/(b+T_a)} \,,$$





with coefficients $a = 7.5$ and $b = 235.0$ for $T_a \geq 0°C$, and $a = 9.5$ and $b = 265.5$ for $T_a < 0°C$.

Actual vapor pressure $e$ (hPa) was then determined as

$$e = e_{\text{sat}} \cdot \frac{rH}{100\%} ,$$

with relative humidity $rH$ in percent, and converted to absolute humidity $\rho_v$ (kg m$^{-3}$) with

$$\rho_v = \frac{e}{T_a + 273.15} \cdot \frac{p}{p - e} \cdot \frac{100}{R_v} \approx 0.217 \cdot \frac{e}{T_a + 273.15} ,$$

with $p$ atmospheric pressure (hPa), and $R_v$ the gas constant for water vapor (461.53 J kg$^{-1}$ K$^{-1}$).

## 2.4 Statistical analyses

Statistical analyses were performed with R version 3.5.2 (R Core Team, 2018). Trend analyses were performed for both trend
in CH$_4$ concentration and drift of TGS 2600 measurements using the Mann-Kendall trend test implemented in the rkt package
that is based on Marchetto et al. (2013). The annual linear trend (or drift) was calculated using the robust Theil-Sen estimator
Akritas et al. (1995) using weekly median values, and the significance of the trend (or drift) was assessed using Kendall's Tau
parameter. All trend and drift estimates were significant at p < 0.05. The highest two-sided p-value of the presented results was
p = 0.000054 and thus no detailed information on p-values is given when statistical significance of trends or drift is mentioned
in the following.

For assessing the quality of the proposed calculation of CH$_4$ concentrations from TGS 2600 sensors we inspected weekly
aggregated data using four key indicators:

**Bias** – the mean of the difference of each 30-min averaged pair of CH$_4$ concentrations in ppm, CH$_{4,\text{TGS}}$ − CH$_{4,\text{ref}}$;

**Stability** – the bias expressed as a percent deviation from the reference CH$_4$ concentration, (CH$_{4,\text{TGS}}$ − CH$_{4,\text{ref}}$)/CH$_{4,\text{ref}}$·100%;

**Variability** – the mean relative deviation of the 95% confidence interval (CI) observed with the TGS 2600 sensor from the
corresponding 95% CI of the CH$_4$ reference measurements (in percent), (CI$_{95\%,\text{TGS}}$ / CI$_{95\%,\text{ref}}$ − 1) ·100%;

**Correlation of median diel cycles** – Pearman's product-moment correlation coefficient between hourly-aggregated median
diel cycles of CH$_4$ measured by the TGS 2600 and reference instruments.

In addition to conventional linear model (least square method) fits we used an artificial neural network approach. This was
performed in Python 3.7.1 using MLPRegressor from sklearn.neural_network version 0.20.2 (Pedregosa et al., 2011). We used
a network with four hidden layers of size 500, 100, 50 and 5, respectively, and an adaptive learning rate. Learning was done with
the data obtained during the calibration period from first measurements until 31 March 2017 and the remaining data (1 April
2017 to 31 December 2018) were used for validation.

## 3 Results and discussion

CH$_4$ concentrations estimated from TGS 2600 measurements during the cold seasons differed strongly from the reference
measurements when the Eugster and Kling (2012) approach was used (not shown); that approach translated the information





from the technical specifications of the TGS 2600 sensor (Figaro, 2005b, a) to outdoor applications. With temperatures above freezing the agreement with the $CH_4$ reference measurements was within $\pm 0.1$ ppm (Fig. 2), but not so during cold conditions ($Ta < 0°C$). The differences between TGS estimates and $CH_4$ reference were largest with the Eugster and Kling (2012) approach

when relative humidity was between 50 and 90% (Fig. 3a). When converting relative humidity to absolute humidity, the results became satisfactory for higher absolute humidity values $> 0.004$ kg m$^{-3}$ (Fig. 3b). Because absolute humidity $> 0.004$ kg m$^{-3}$ is only possible at temperatures $> 0°C$ it appears quite obvious that temperature and humidity corrections of solid state sensors most likely do not relate to relative humidity (which is a ratio and not a physical variable of atmospheric water content), but to either actual vapor pressure (in hPa) or absolute humidity (in kg m$^{-3}$). In all tested models absolute humidity performed

marginally better than vapor pressure (measured by R$^2$, not shown), hence we suggest the following model and parameterization to estimate $CH_4$ concentrations in ppm from TGS 2600 signal voltage measurements:

$$
\begin{aligned}
CH_4 \quad = \quad & 1.425 + 0.12\,S_c + 0.375/S_c - 0.0065\,T_a + \qquad\qquad (2)\\
& + 53.3\,\rho_v + 0.0022\,S_c \cdot T_a - 0.0017\,T_a/S_c + \\
& + 4.9\,S_c \cdot \rho_v - 67.4\,\rho_v/Sc - 0.39\,S_c \cdot T_a \cdot \rho_v \\
& + 1.15\,T_a \cdot \rho_v/S_c \,,
\end{aligned}
$$


with $S_c$ the dimensionless sensor signal (see Eq. 1), $T_a$ ambient air temperature in °C, and $\rho_v$ absolute humidity in kg m$^{-3}$. The parameter estimates were derived from the entire dataset 2012–2018 for TGS sensor #1 (Table 1, "entire period"). For other sensors the result from Eq. (2) can be considered as a linearized signal that can be fine-tuned with a sensor-specific two-point calibration as suggested in Section 3.4 of Eugster and Kling (2012).

## 3.1  Performance of the TGS 2600 sensor at 30-minute resolution

Using Eq. (2) yields satisfying agreement with 30-minute averaged data under both typical low Arctic summer and winter conditions (Fig. 4) with an overall R$^2$ of 0.42 (Table 1). A detailed inspection of four representative seven-week time periods at full 30-minute resolution is shown in Figures 5–8. Typical summer conditions at the beginning of this study (Fig. 5) and towards the end of the analyzed period (Fig. 6) indicate that the short-term agreement (R$^2$ = 0.65, Fig. 5) is better when the

TGS sensor was still relatively new than at age seven (R$^2$ = 0.38, Fig. 6), but the variability decreased from –42% to –9% with no relevant difference in bias and stability (0.01 ppm and 0.4% vs. 0.00 ppm and 0.0%, respectively). In winter the timing of most events is correctly captured (Fig. 7) with a R$^2$ of 0.45, but the dynamics are not satisfactorily captured by the TGS sensor, indicated by a 59% underestimation of the 95% CI during this mid-winter period. The transition from warm to cold season (Fig. 8) shows a mixture of days, where the regular diel cycle, which is typical for the warm season, is still adequately

captured, but the dynamics of periods with air temperature $<0°C$ (see Fig. 4) when $CH_4$ concentrations tend to be highest as in winter (Fig. 7), are not adequately captured. Still, with a R$^2$ of 0.51 (Fig. 8) more than 50% of the variance observed in the 30-minute averaged $CH_4$ reference measurements is captured by the low-cost TGS 2600 sensor.


### 3.2 Performance of weekly aggregated data

The TGS 2600 is not expected to provide short-term accuracy comparable to high-quality instrumentation (see also Lewis et al.,
2018). However, Eugster and Kling (2012) argued that such measurements still may provide additional insights as compared to
the passive samplers described by Godbout et al. (2006a, b), integrating over longer time frames. Thus, here we inspected the
performance of weekly aggregated estimates derived from the TGS 2600 in order to inspect drift of the two sensors and their
performance over the seven-year deployment period. Note, that in Eq. (2) we did not include a drift correction. Figure 9 shows
weekly medians of sensor signals, the difference between the raw signals from both sensors mounted at the same position (Fig.
1), and the agreement with the reference signal. The two TGS 2600 sensors (#1 and #2) showed a trend in their signals of
–18.8 mV yr$^1$ and –15.5 mV yr$^{-1}$, respectively (Fig. 9a,b). Thus, with typical signals on the order of 200–700 mV (Fig. 9a)
the lowest (winter) readings may no longer be measurable after 10–13 years of continuous operation, indicating the end of life
of a TGS 2600.

Figure 10 shows the weekly median bias, variability, and the correlation between the weekly aggregated median diel cycle
of $CH_4$ at hourly resolution between the TGS #1 measurements and the reference. Despite the trend of the sensor signal shown
in Figure 9a,b both the bias and variability primarily show a seasonal pattern with a slightly negative bias (around –0.02 ppm)
during peak growing season and a corresponding positive deviation in mid-winter when temperatures can be well below –30°C
(Fig. 10a). The variability (Fig. 10b) shows the inverse pattern of the bias. If bias is expressed as the relative bias (i.e., stability),
the stability vs. variability plot (Fig. 11) shows points lying uniformly around the line of a –1:1 relationship ($R^2 = 0.67$).

### 3.3 Linear trend and drift estimates

All linear trend estimates were statistically significant (see Section 2.4). However, our measurements started with warm-season
measurements only (2012–2014) that were successively expanded to include cold season measurements. Thus, all interpre-
tation of the trends and drifts presented here should be considered with caution given the long gaps in data due to technical
challenges operating such equipment under adverse winter conditions. The $CH_4$ concentration trend observed with the high-
quality reference measurements was 10.1 ppb yr$^{-1}$. This is 2.5 times the trend observed from 2005–2011 by NOAA (28.6 ±
0.9 ppb or 4.09 ppb yr$^{-1}$; Hartmann et al. (2014); their Table 2.1), but of the same order of magnitude reported by Nisbet
et al. (2014) for 2013 (last year covered by that study) for latitudes north of the Tropic of Cancer. Thus, this trend may be real
and hence all trends seen in low-cost sensor signals are not necessarily solely an artefact of such sensors. It however remains
a challenge to deduce the true trend in $CH_4$ concentrations over longer time periods using such a low cost sensor because
of drifting signals. Thus, we inspected the drift of the TGS 2600 derived concentration with respect to the (true) $CH_4$ trend
observed with the high-quality reference instrument. These drifts appear to be smaller than the true trend, but are still consider-
able: the bias of TGS-derived $CH_4$ concentrations drifted by 4–6 ppb yr$^{-1}$ (40–60% of actual trend) and variability drifted by
–0.24% yr$^{-1}$. They provide encouraging results suggesting that with occasional (infrequent) calibration against a high-quality
standard, e.g. using a traveling standard operating during a few good days with adequate coverage of the near-surface diel cycle
of $CH_4$, TGS 2600 measurements might be suitable for the monitoring of $CH_4$ concentrations also in other areas. As shown





in Figure 10c the correlation of median diel cycles between TGS estimates and $CH_4$ reference is one of the weak points in the current performance of the TGS 2600 sensors. Also, we observed a significant negative trend of the correlation coefficient of $-0.051$ yr$^{-1}$ (Fig. 10c). However, the key finding is that the typical diel cycle during the warm season (air temperature $> 0°C$) disappears during winter conditions (Figs 4, 7) and thus separate transfer functions for warm and cold temperatures might be

a solution for future studies (Table 1). Our Eq. (2) is thus on purpose derived from the entire dataset to provide a starting point for more elaborate fine-tuning in projects where this is desired.

### 3.4 Potential of using artificial neural networks

Casey et al. (2019) found that artificial neural networks (ANNs) outperformed linear models in mitigating curvature and linear trends in trace gas measurements when used with the same set of input variables during a three-month comparison period.

To inspect the potential of ANNs at our Arctic long-term dataset, we added the ANN results to Figures 5–8. In summer (Figures 5, 6) we did not find a substantial advantage of ANN over the linear approach of Eq. (2) in terms of root mean square error (RMSE) of the prediced $CH_4$ concentrations (Table 1). In particular, the ANN was unable to better reflect the skewed distribution of diel variations of $CH_4$ concentrations and tended to underestimate daytime minimum concentration without a clear improvement of the fit to nocturnal peaks. It should be noted that at this latitude the sun does not set between 24 May and

20 July, thus nocturnal conditions are clearly different from conditions at lower latitudes such as the ones investigated by Casey et al. (2019). Similarly, the transition from warm to cold season (Fig. 8) does also not provide strong evidence of improvements with an ANN that uses the same input variables as a linear model. However, in winter when the diel cycle of all variables is less dominant than during the warm and transition seasons, the ANN approach is capable of producing a more realistic $CH_4$ estimate from TGS 2600 signals than the linear model approach (e.g., Fig. 7). This is not very surprising as the prediction range

of a linear model decreases linearly with the amplitude of the input variables (e.g., in absolute humidity, which does not vary strongly at sub-zero temperatures), whereas an ANN does not have such linearity constraints.

The drawback of ANNs is the difficulty in determining why the fit has become better than an empirical linear model fit. On the other side, this could encourage future work to better understand the physical response of TGS sensors to cold environments. In summary, we strongly recommend to focus on physical relationships, because it is unclear what scientific insights an ANN

fit could offer in addition to the technically nicer fits to data.

### 3.5 Suggestions for future work

The residuals of Eq. (2) are homoscedastic both when plotted against air temperature (Fig. 12a) or absolute humidity (Fig. 12b). Deviations are generally constrained within $\pm0.1$ ppm or better, but with higher variability at both temperature ends where data coverage is poor (gray bars at bottom of Fig. 12a) as temperatures $< -30°C$ were not frequently covered due to

technical problems with the measurement station, or at summer temperatures $> 20°C$ that are still rather rare at this low Arctic latitude (Hobbie and Kling, 2014). A slightly different picture emerges for low absolute humidity: 56% of measurements are at lower humidities than the saturation humidity at $0°C$ ($0.0049$ kg m$^{-3}$), thus the rather homgenous variances at low humidity





(Fig. 12b) indicate that humidity is not of concern at low temperatures, and future attempts for improvements should rather focus on humidity $> 0.01$ kg m$^{-3}$ and temperatures $> 20$°C that are not normally found in the Arctic.

Another approach was taken by van den Bossche et al. (2017) who performed an in-depth laboratory calibration of the very similar but less sensitive Figaro TGS 2611-E00 sensor (Figaro, 2013, the manufacturer only shows a response above 300 ppm CH$_4$) at different temperatures and relative humidity over a CH$_4$ calibration range starting at $\approx 2$ ppm ambient concentration up to 10 ppm CH$_4$. Despite the effort, the residual concentrations remained large (range of ca. –1.5 ppm to +1.1 ppm) – too large for the application we present here. Our efforts to calibrate our TGS 2600 sensors in a laboratory climate chamber in a

similar way was not satisfactory (Eugster, unpublished), hence our approach presented here to determine the sensor behavior from long-term outdoor measurements under real-world conditions. Contrastingly, Kneer et al. (2014) are convinced that "to be of use for advanced applications metal-oxide gas sensors need to be carefully prepared and characterized in laboratory environments prior to deployment". We however do not fully agree because laboratory conditions simplify the real world too much, and it is difficult to carry out laboratory treatments from –41°C to 27°C as it would be required for our Arctic site. The

data we present indicate that most likely it is absolute humidity, not relative humidity, that should be used for such calibrations, which in principle should provide the best quality results if the relevant factors are known and can be included in the calibration set-up.

Using absolute humidity in place of relative humidity for the correction of the TGS 2600 was already attempted by Collier-Oxandale et al. (2018), which contrasts with the manufacturer's suggestion Figaro (2005a). However, as mentioned above,

relative humidity is not a physical entity and hence using actual vapor pressure or absolute humidity may indeed be a better approach to understand the TGS 2600 sensor sensitivity to environmental conditions other that CH$_4$ concentrations.

The TGS 2600 sensor's best performance is in applications where passive samplers would be another option (see also Eugster and Kling, 2012). Contrastingly, using the TGS 2600 for short-term measurements (resolution of seconds to minutes) has not yet led to satisfactory results (Kirsch, 2012; Falabella et al., 2018). In our dataset we found that adding wind speed to the

calculation model slightly improved the model fit during the warm season, but because no reliable continuous winter wind speed measurements were possible at the TWE site we did not include wind speed in our Eq. (2). However, this may be a key component for understanding the variability of TGS 2600 measurements when flying an unmanned aerial vehicle (UAV) where turbulent conditions may change within seconds to minutes. To address this additional factor, we produced a heat loss model, assuming that the sensor correction is related to the cooling of the heated surface of the solid state sensor, which has a nominal

surface temperature $T_s = 400$°C (Falabella et al., 2018) that is the typical operation temperature of SnO$_2$-Ni$_2$O$_3$ sensors (Hu et al., 2016). Our candidate model for heat loss ($HL$ in W) was

$$HL \sim \xi \cdot \overline{u}^2 \cdot (T_s - T_a) \cdot (\rho_d \cdot C_d + \rho_v \cdot C_v) \; , \tag{3}$$

with $\overline{u}$ mean horizontal wind speed (m s$^{-1}$), $T_s$ and $T_a$ sensor surface and ambient air temperature (K), respectively, $\rho_d$ density of dry air (kg m$^{-3}$), $\rho_v$ absolute humidity (kg m$^{-3}$), and $C_d$ and $C_v$ heat capacity of dry air and water vapor, respectively (J

kg$^{-1}$ K$^{-1}$). The scaling coefficient $\xi$ is a best fit model parameter (units: s m). The assumption made here was that the wind speed governs the eddy diffusivity of heat transported along the temperature gradient between the sensor surface and ambient





air, and the moisture correction is only associated with the fact that water vapor has a higher heat capacity (1859 J kg$^{-1}$ K$^{-1}$) than dry air (1005.5 J kg$^{-1}$ K$^{-1}$), and hence the heat capacity of moist air increases accordingly with $\rho_v$. However, although this approach is more mechanistic than Eq. (2), it's ability to predict CH$_4$ concentration from TGS 2600 measurements was

much worse than that of the empirical linear model and ANN approaches (Table 1). But in order to make further progress on improving the transfer function from TGS 2600 signals to defensible CH$_4$ concentrations it will be essential to increase our understanding of the physical processes that influence such measurements. This is not an easy task since there is a substantial proprietary knowledge that is unrevealed by the manufacturer. Newer, promising developments are underway that work with a mixed potential sensor using tin doped indium oxide and platinum electrodes in combination with yttria-stabilized zirconia

electrolyte that show a logarithmic signal range of 0–10 mV for the range of 1–3 ppm CH$_4$ of interest for ambient air studies (Sekhar et al., 2016). The basic principle that the active metal-oxide is charged with O$_2$ (or O$^{2-}$), which then oxidizes CH$_4$, seems to be similar to the SnO$_2$-based TGS 2600; thus there is a good chance that our findings for the TGS 2600 are also useful for assessing the performance of newer solid-state sensors with different active materials.

## 4   Conclusions

We present the first long-term deployment of two identical TGS 2600 low-cost sensors that show a (weak) sensitivity to ambient levels of CH$_4$ (here: range 1.824–2.682 ppm as measured by a high-quality Los Gatos Research reference instrument). We suggest a new transfer function to correct the TGS 2600 signal for cross-sensitivity to ambient temperature and humidity that also yields satisfactory results under cold climate (Arctic) conditions with temperatures down to –40°C. This was only possible by using absolute humidity and not relative humidity for the correction. With this correction determined over the

entire 2012–2018 data period, the 30-minute average CH$_4$ concentration could be derived from TGS 2600 measurements within ±0.1 ppm. The two completely different regimes of diel CH$_4$ concentration variations during the cold season (typically with a snow cover and frozen surface waters) and the warm season (when plants are active in the low Arctic) suggest that further improvements can be obtained by more specifically developing separate transfer functions for cold and warm conditions. The use of an artificial neural network shows the potential for improvements in transfer functions under very cold conditions.

We consider the quality of TGS 2600 derived CH$_4$ estimates adequate if aggregated over reasonable periods (e.g., days or one week), but caution should be taken with application where short-term response is of key relevance (e.g., within seconds to minutes required by mobile measurements with UAVs). The deterioration of the sensor signal over time indicates that a TGS 2600 operated under ambient conditions as in our deployment at a low Arctic site in northern Alaska (Toolik wet sedge site) has an estimated life time of ca. 10–13 years. Thus, there is a potential beyond preliminary studies if the TGS 2600 sensor is

adequately calibrated and placed in a suitable environment where cross-sensitivities to gases other than CH$_4$ is of no concern.

*Data availability.* The data used in this study can be downloaded from the ETH Zurich Research Collection via doi:10.3929/ethz-b-000369689 [note that the dataset will be uploaded after acceptance of the paper, thus the reserved doi is not yet active]



*Author contributions.* WE, JL and GWK designed the study, set up the instrumentation, and serviced the site. WE carried out the main analyses. JE helped with the main analyses, set up and carried out the ANN calculations. WE wrote the manuscript and all co-authors

worked, commented, and revised various versions.

*Competing interests.* None.

*Disclaimer.* The authors are independent from the producers of the instruments and sensors referenced in this article, and thus the authors do not have a commercial interest to promote any of the mentioned products.

*Acknowledgements.* We thank Jeb Timm, Colin Edgar, and other members of the Toolik Field Station science support staff for field help

under difficult conditions. We also thank support staff from CPS for help with power supplies, technicians and students supported on several NSF grants, as well as several students supported by the NSF-REU program for help in the field over the years.

We acknowledge support received from the Arctic LTER grants NSF-DEB-1637459, 1026843, 1754835, NSF-PLR 1504006, and supplemental funding from the NSF-NEON and OPP-AON programs. Gaius R. Shaver (MBL) is acknowledged for initiating the study and supporting our activities in all aspects.



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





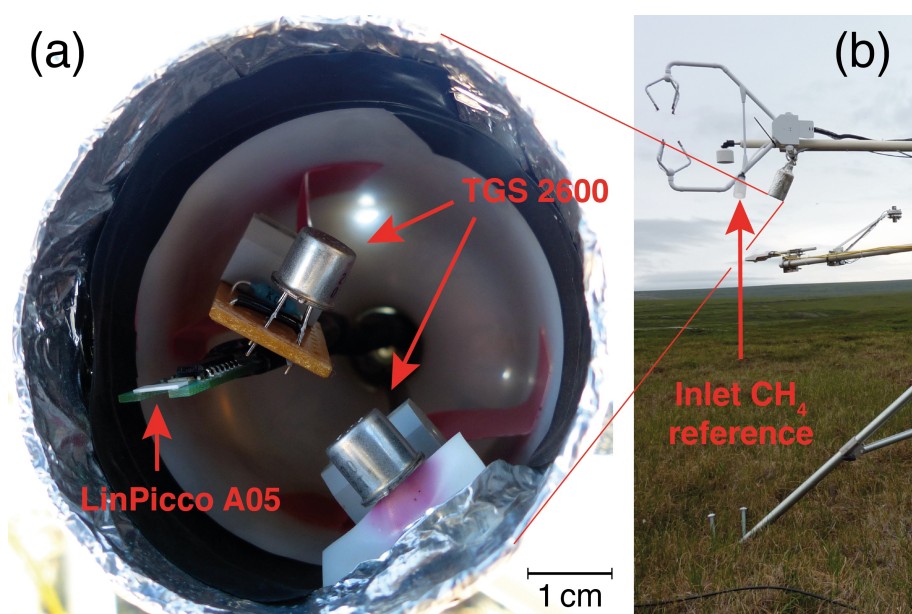

**Figure 1.** Two TGS 2600 trace gas sensors and the LinPicco A05 temperature and relative humidity sensor (a) inside the weather protection, and (b) the mounting position of the TGS weather protection and reference $CH_4$ gas inlets at the Toolik wet sedge eddy covariance flux site.




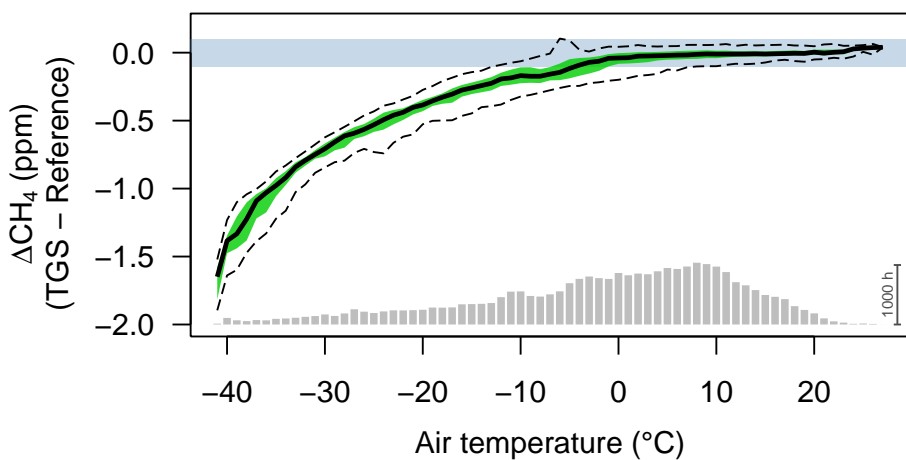

**Figure 2.** Difference between TGS2600 and reference $CH_4$ measurements (30-minute averages) as a function of air temperature when using the Eugster and Kling (2012) conversion. Agreement was good when ambient temperature was above freezing. The horizontal color bar shows the $\pm 0.1$ ppm range around a perfect agreement. The green band shows the inter-quartile range of bin-averaged differences (Figaro sensor #1), and dashed lines show the extent of the 95% confidence intervals. Gray bars at the bottom show number of 30-minute averages in each bin. The scale bar (1000 hours) at right specifies their size.



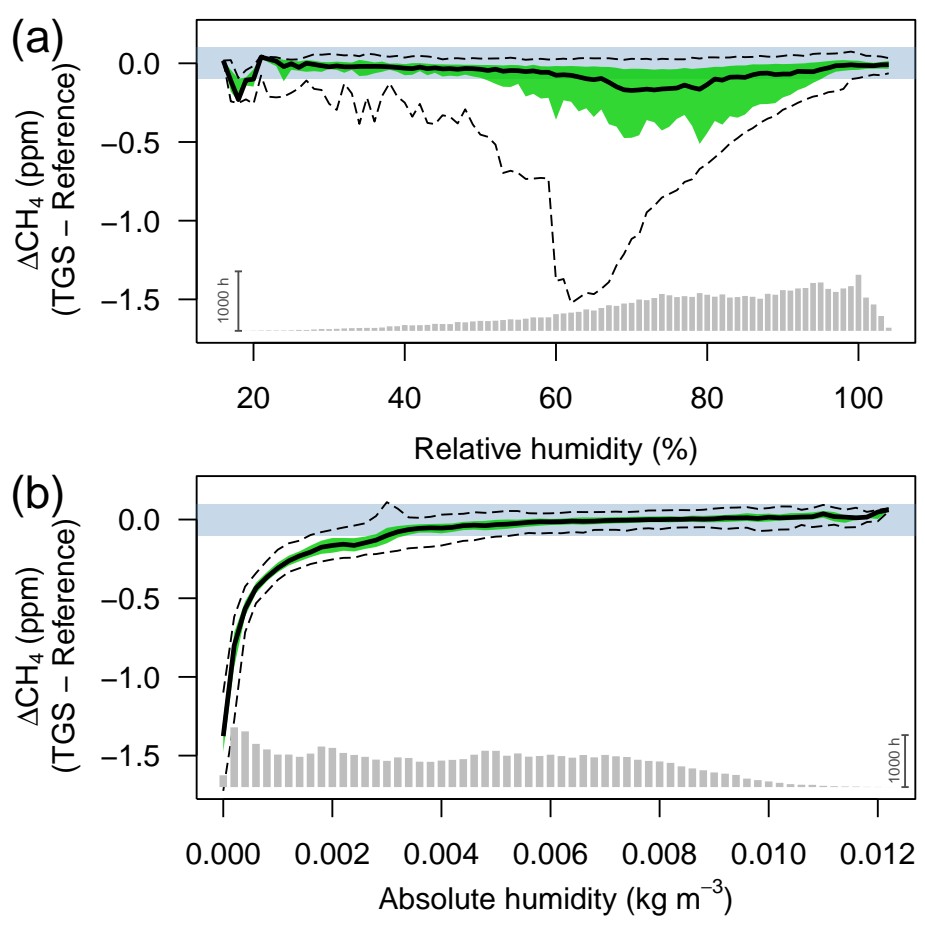

**Figure 3.** Difference between TGS2600 and reference $CH_4$ measurements (30-minute averages) as a function of (a) relative humidity (in %) and (b) absolute humidity (in kg m$^{-3}$). The horizontal color bar shows the $\pm 0.1$ ppm range around a perfect agreement. The green band shows the inter-quartile range of bin-averaged differences (Figaro sensor #1), and dashed lines show the extent of the 95% confidence intervals. Gray bars at the bottom show number of 30-minute averages in each bin. The scale bar (1000 hours) at right specifies their size.





**Figure 4.** Overview over annual courses of 30-minute averaged air temperature (green) and CH$_4$ concentrations (blue and red). Pale color bands show the daily inter-quartile range (50% of values between first and third quartile) of measurements from all years. Solid lines show actual measurements. Red lines are the reference CH$_4$ measurements, and blue lines show the CH$_4$ concentration derived from TGS 2600 measurements. Actual measurements show 30-minute mean values.



**Figure 5.** Timeseries (a) of TGS #1 derived $CH_4$ during a 7-week snow- and ice-free period in the first year of the long-term deployment (2012), (b) correlation with reference concentration, and (c) residuals (TGS 2600 – Reference) of 30-minute averaged measurements. Thin solid lines in (a) show the result when all data are used with Eq. (2); reference concentration is shown with a red bold line; c/v shows an alternative fit from splitting the available data into a calibration and a validation part; and the dashed line shows the performance of an artificial neural network (ANN) fit.

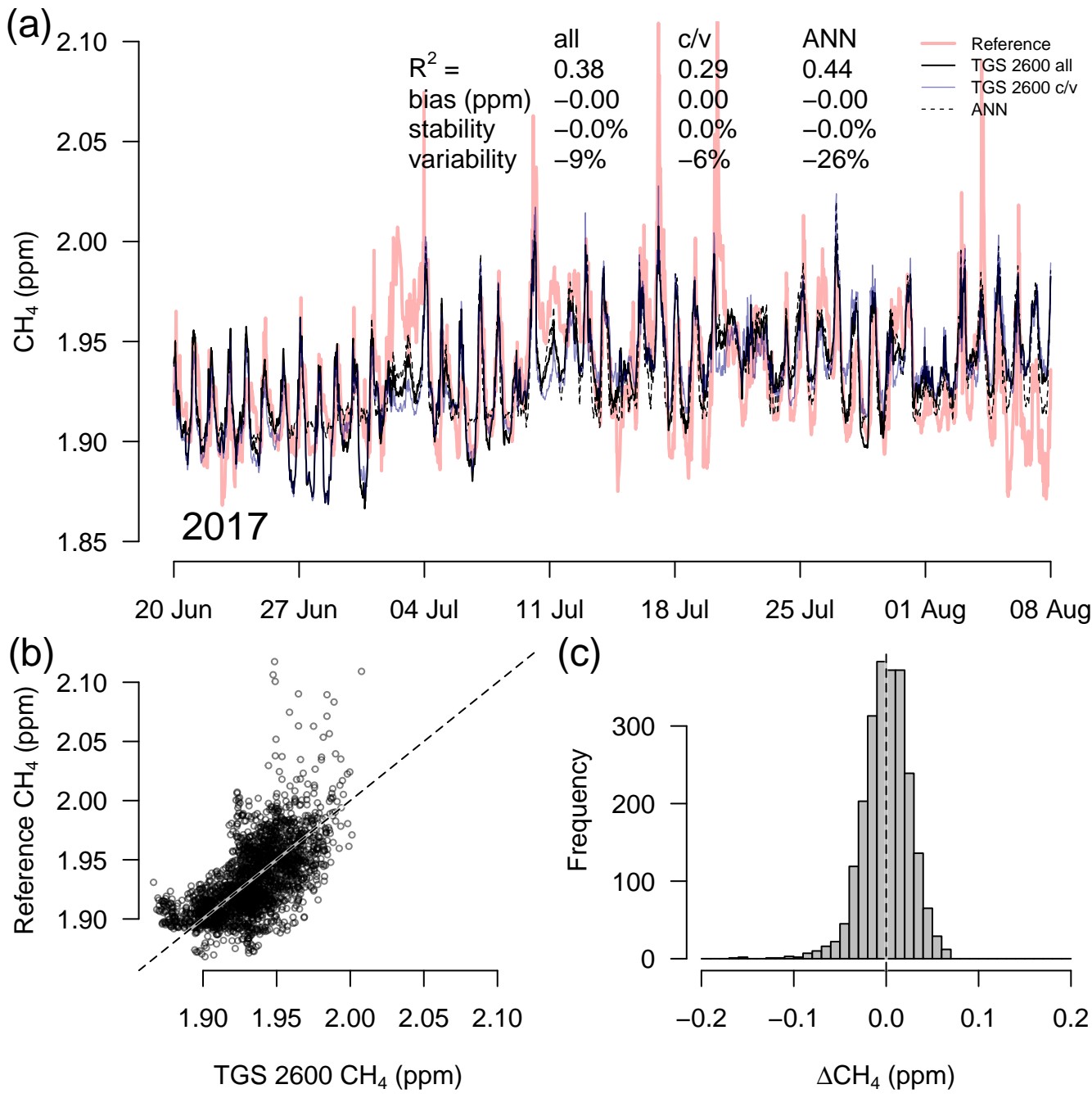

**Figure 6.** As in Figure 5 but with measurements from a 7-week snow- and ice-free period in 2017 at sensor age of seven years.







**Figure 7.** As in Figure 5 but with measurements from a 7-week period in mid winter with temperatures plunging down to –40°C. High CH$_4$ concentrations coincide with the coldest temperatures (see Fig. 4).





**Figure 8.** As in Figure 5 but with measurements from a 7-week period during the transition from fall to early winter.

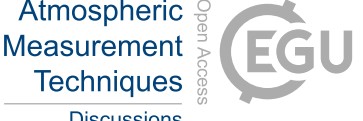

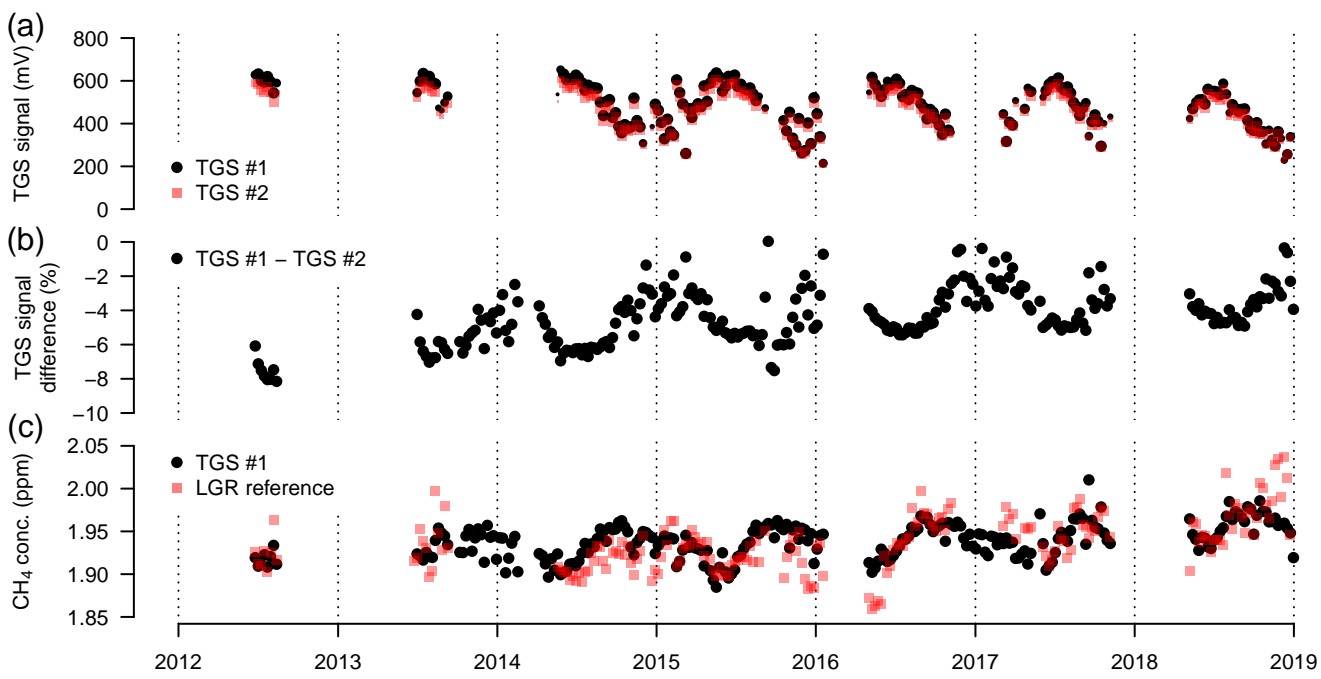

**Figure 9.** (a) Weekly median sensor signals from both TGS 2600 sensors, (b) relative signal difference between the two TGS 2600 sensors in percent of the signal measured with the primary sensor #1, and (c) $CH_4$ derived with Eq. (2) for TGS sensor #1 and measured by the Los Gatos Research reference instrument. Symbol size is proportional to relative data coverage.



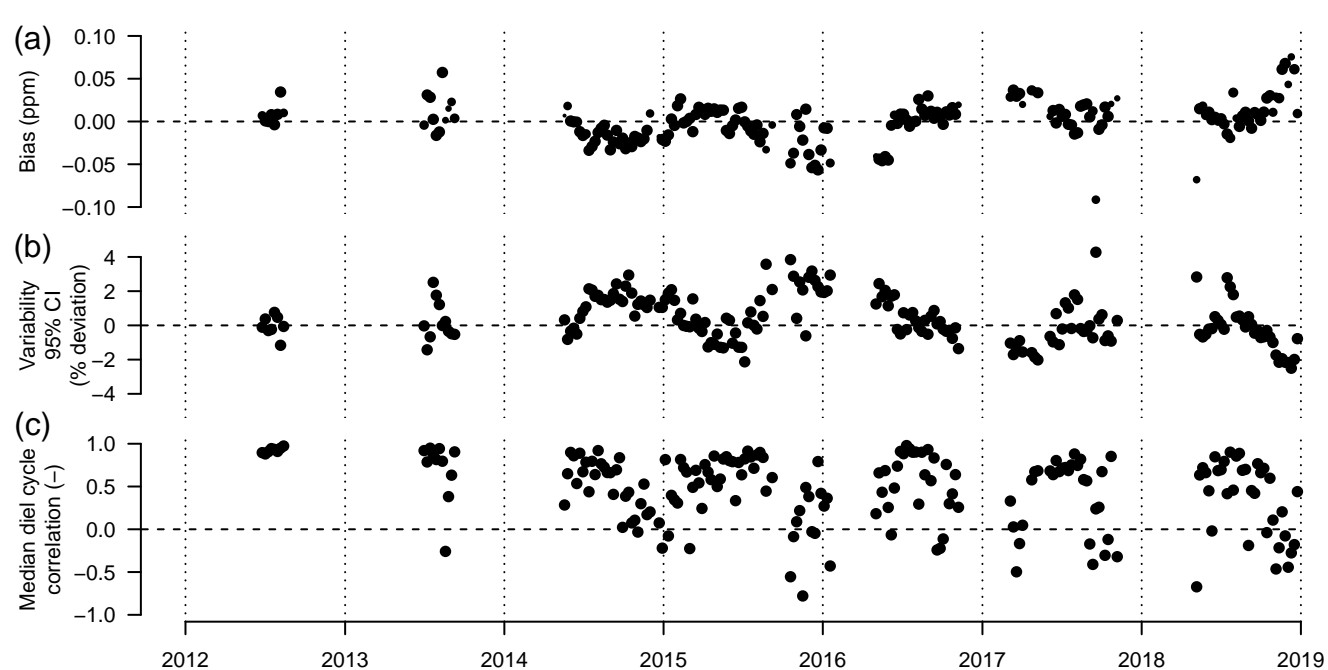

**Figure 10.** (a) Weekly median bias, (b) median variability, and (c) correlation between weekly median diel cycles of TGS 2600 sensor #1. Symbol size is proportional to relative data coverage.





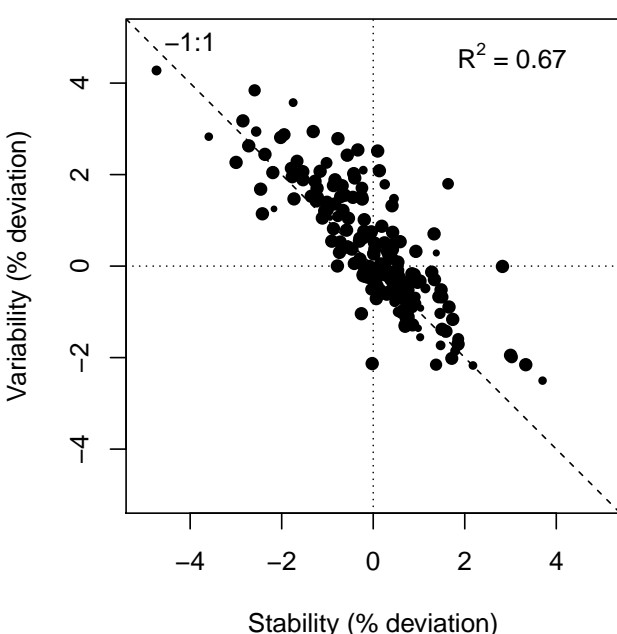

**Figure 11.** Variability and stability (relative bias) of weekly median TGS 2600 sensor #1 are inversely related and plot along the −1:1 line. Symbol size is proportional to relative data coverage.

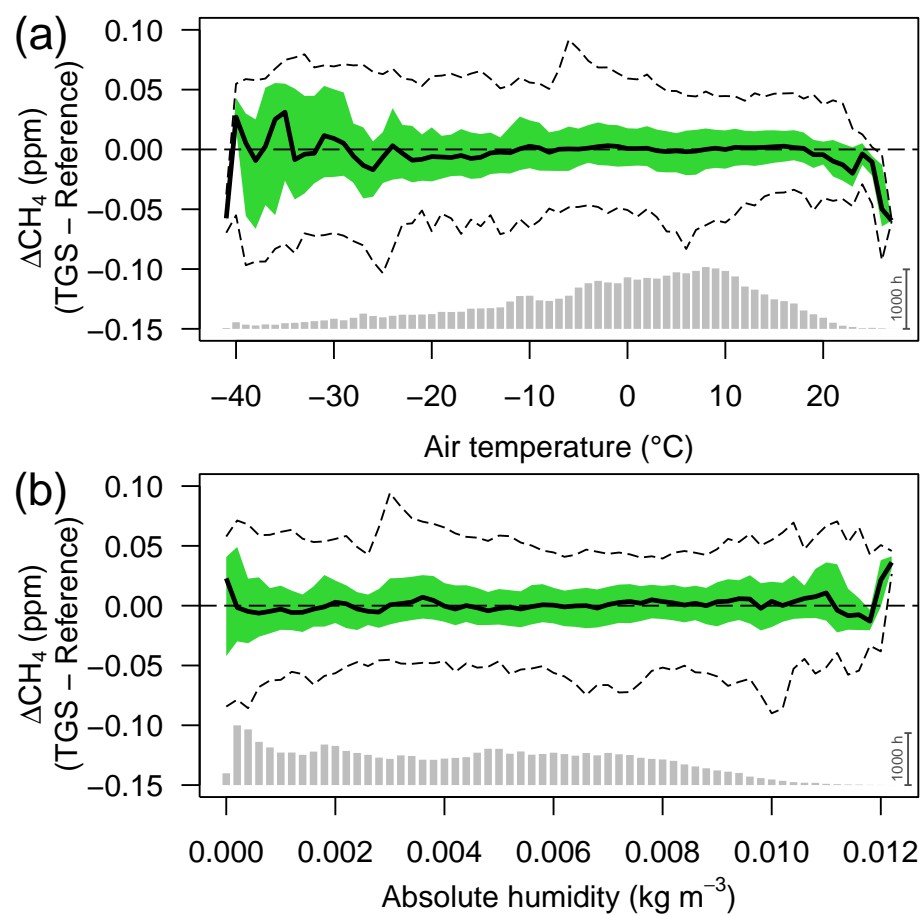

**Figure 12.** Residuals of 30-minute averaged TGS 2600 vs. reference $CH_4$ measurements as a function of (a) air temperature, and (b) absolute humidity. Colored areas show the inter-quartile range (50% CI), bold lines show the median, and dashed lines show the bin-averaged 95% confidence interval. Bin size was 1°C and 0.0002 kg m$^{-3}$, respectively. Gray bars at the bottom show number of 30-minute averages in each bin; the scale bar (1000 hours) at right specifies their size.



**Table 1.** Goodness of fit of TGS 2600 derived $CH_4$ concentrations (30-minute averages) obtained from a linear model using air temperature and absolute humidity (Eq. 2), a heat loss model (Eq. 3), and an artificial neural network (ANN). For the goodness of fit the coefficient of determination ($R^2$) and the root mean square error of the residuals (RMSE) are reported for the overall model and separately for warm and cold conditions. The parametrization of the linear model given in Eq. (2) used the entire period 2012–2018. For a more rigorous model test, all three approaches were calibrated with the data measured in years 2014–2016, and the remaining data (2012–2013 and 2017–2018) were used for validation.

| | Linear Model | | | Heat Loss Model | | Artificial Neural Network | |
|---|---|---|---|---|---|---|---|
| | Entire period | Calibration | Validation | Calibration | Validation | Calibration | Validation |
| *Overall* | | | | | | | |
| $R^2$ | 0.42 | 0.45 | 0.21 | 0.17 | 0.28 | 0.51 | 0.54 |
| RMSE (ppm) | 0.03 | 0.03 | 0.04 | 0.03 | 0.05 | 0.03 | 0.03 |
| *Warm conditions ($T_a \geq 0°C$)* | | | | | | | |
| $R^2$ | 0.48 | 0.52 | 0.29 | 0.18 | 0.18 | 0.56 | 0.47 |
| RMSE (ppm) | 0.03 | 0.03 | 0.03 | 0.03 | 0.04 | 0.03 | 0.03 |
| *Cold conditions ($T_a < 0°C$)* | | | | | | | |
| $R^2$ | 0.32 | 0.34 | 0.03 | 0.16 | 0.05 | 0.44 | 0.37 |
| RMSE (ppm) | 0.03 | 0.03 | 0.05 | 0.03 | 0.05 | 0.03 | 0.03 |