# Peer review of "Long-term reliability of the Figaro TGS 2600 solid-state methane sensor under low Arctic conditions at Toolik Lake, Alaska"

_Atmospheric Measurement Techniques, 2019_

## Referee Comment (RC1) · Anonymous Referee #1 · 17 Jan 2020

**1   General comments**

The manuscript summarises the findings from a 6-year field deployment of a small, low-cost methane sensor under low Arctic conditions. Given the interest of the community in small, low-cost sensors on the one hand and the measurement of atmospheric methane on the other hand, this work is of high relevance to the readers of Atmospheric Measurement Techniques.

The manuscript is written and structured well. The methods are described in appropriate detail. However, I see some shortcomings in the data analysis and presentation of

results, detailed below, that should be addressed before publication.

**2  Specific comments**

It is unclear which quantity the Authors use when they report the abundance of methane. The sentence 'We report all gas concentrations in mixing ratios by volume (ppm or ppb dry mole fractions)' (ll. 78–79) is contradictory, as it uses three distinct quantities as if they were synonymous. In a mixture of two components A and B, concentration $c$ is defined as $c = q_A/(V_A + V_B)$ , where $V$ is volume and $q$ one of the quantities mass, amount, volume or number concentration [1]. Mixing ratio by volume $r$ is rather uncommon and defined as $r = V_A/V_B$ [2]. Mole fraction $x$ (IUPAC recommends $y$ for gaseous mixtures, but this is not common in atmospheric science) finally is defined as $x = n_A/(n_A + n_B)$, where $n$ is amount of substance [3]. Given that the WMO scale for methane abundance is a mole fraction scale [4], the reporting of mole fractions would be desirable. While the use of the term 'concentration' for mole fraction is accepted for communication with the general public [4], a publication in a scientific journal should in my eyes favour exact terminology. In any case the Authors must make clear which quantity is reported.

Two TGS 2600 sensors were deployed at the site, referenced to as #1 and #2. However, in several instances in the manuscript 'TGS 2600' appears without a number when I think TGS 2600 #1 is meant. Also, 'sensor', 'TGS' and 'TGS2600' are used. This should be made more consistent. Results from TGS 2600 #2 are presented exclusively in l. 104, ll. 185–186 and Fig. 9. Explaining the minor role of TGS 2600 #2 around l. 59, l. 104 or l. 162 might prevent confusion of the reader.

ll. 74–77     How often and to which scale were the reference analysers calibrated?

l. 153       '[...] relative humidity (which is a ratio and not a physical variable of atmospheric water content)' – I think I have an idea of what the authors mean, but I find the wording not quite right. Would the authors say that the refractive index of a material is not a physical variable because it is a ratio? In general, I miss some thoughts about the temperature dependence of the quantity used for expressing humidity. The Figaro TGS 2600 has a heated sensing element, so the relative humidity at the sensing surface is different from the relative humidity in the environment. The temperature dependence of relative humidity makes this quantity a less than ideal choice for this type of correction. Unfortunately, both alternative quantities chosen by the authors to express water vapour content depend on temperature as well. Mixing ratio (by mass) or specific humidity would be temperature-independent alternatives [6]. Using the ideal gas equation, the terms in Eq. 2 that contain the product $T_a \cdot \rho_v$ would also cancel out the temperature dependence of absolute humidity if $T_a$ was absolute temperature (in K) – but in the manuscript a Celsius temperature is used. Hence, my suggestion to the authors is to try out either mixing ratio by mass or specific humidity as an independent variable in Eq. 2. Using absolute instead of Celsius temperature might be advisable as well.

ll. 161–162  Using the entire dataset for estimating the parameters of Eq. 2 is a comprehensive test of how well the model can describe the dataset, but is of limited relevance for field deployments where calibrations are performed during limited periods of time and the main interest is in the uncertainty of independent measurements. For this reason, splitting the dataset into a calibration and a validation part yields important insights. The caption of Table 1 explains that the authors have in fact performed analyses of a split data set. This fact should also be mentioned in the main text around the lines given.

l. 167  For the reasons given before, the results presented in the columns 'Linear

Model - Calibration' and 'Linear Model - Validation' in Table 1 should be discussed here, even more so because the results for the validation period are substantially worse than for the period used for calibration.

ll. 193–194  Is there any conclusion that can be drawn from this finding of a -1:1 relationship?

Sect. 3.4  The discussion in Sect. 3.1 leans heavily on the coefficient of determination $R^2$. In Table 1, each $R^2$ for the ANN approach is higher than the corresponding $R^2$ for the Linear Model. Considering just the validation period, the ANN approach outperforms the Linear Model by a factor of 3–10 by this measure. Similarly, in Fig. 6 and 8 the ANN approach outperforms the Linear Model (comparing $R^2$ of 'ANN' and 'c/v', 'all' is irrelevant in this respect); in Fig. 5 and 7 they perform nearly equally well. None of these comparisons is made here. Instead, the authors state that the root mean square error (RMSE) does not improve substantially with the ANN approach. While I generally appreciate the reporting of RMSE together with $R^2$, its interpretation here is questionable. On the one hand, RMSE is reported in Table 1 with one significant digit only, potentially masking up to ~30% differences for an RMSE of 0.03 ppm (0.025 ppm vs. 0.0349 ppm). On the other hand, the RMSE should be seen in the context of the variability of the data, specifically the root mean square difference between the reference measurements and their mean value over the whole dataset, which is not stated. Overall, the discussion in this section appears negatively biased with regard to the ANN approach. This also manifests in the last paragraph of this section, where 'understand[ing] the physical response of TGS sensors' is prioritised over 'technically nicer fits to data', a stark contrast to the lack of a physical interpretation of the terms in the empirical model (Eq. 2). Section 3.4 must be revised to reach the level of neutrality expected from a scientific publication.

ll. 229–231   To make such an argument, the reader must be informed about the amplitude of all input variables, especially $S_C$.

l. 237        I might be mistaken, but as far as I understand the term homoscedasticity it would in this case mean that the variance of the deviation in CH4 abundance is the same for every temperature bin. The authors do not report variances, but both interquartile range and 95% confidence interval suggest that the variance is higher at low temperatures than at high temperatures, i.e. heteroscedasticity.

ll. 253–257   'laboratory conditions simplify the real world too much' – What could be the simplification that makes laboratory calibrations problematic? The input variables used in the empirical model (Eq. 2) can – practical difficulties taken aside – be controlled in the lab. Any other variable that might prevent transfer of lab results to field conditions is not included in the empirical model, so the problem would not be a simplification of the lab environment but a model deficiency. The last sentence of the paragraph seems to go in this direction ('relevant factors'), but is unclear. Please explain better or leave out.

ll. 258–261   Suggesting to move the first sentence to l. 150 and to remove the other one (repetition).

Fig. 5–8      The graphs are squeezed in horizontal direction, making comparisons between the lines difficult. A shorter period, e.g. 14 days, would give more insight.

Fig. 5 and 7  The collected in 2012 and 2015 are both part of the calibration period, not the validation period, which is important to know for the reader to correctly interpret 'TGS 2600 c/v' and 'ANN'. I therefore strongly suggest a note in the figure caption.

| Fig. 5 | Suggesting to replace '(TGS 2600 - Reference)' with '(TGS 2600 all - Reference)' in the caption |
| --- | --- |
| Fig. 9 | A plot of the difference of the methane abundance calculated from the measurements of the two sensors would be of high interest for the readers. With such a new panel it is also important to state if the parameters derived for TGS 2600 #1 have been used when applying Eq. 2 to the measurements of TGS 2600 #2. In my opinion the new panel could replace panel (b), as the signal difference seems of little relevance. |
| Fig. 10 | If the main text in ll. 189–190 is correct, 'and the reference' is missing at the end of the first sentence of the figure caption. |

**3 Technical corrections**

| l. 1 | Suggesting to remove "weak" to avoid misunderstanding. Alternatively, it could be written in parentheses like it the conclusions. |
| --- | --- |
| l. 8 | Insert a space between value and unit of temperature. This correction is necessary wherever '°C' is used [5]. |
| l. 76 | replace**d** |
| l. 140 | typeset 'Ta' as $T_a$ |
| l. 305 | 'cross-sensitivities [...] **are** of no concern' |
| Fig. 2 | There seems to be a non-displayable glyph at the beginning of the label for the vertical axis, possibly a $\Delta$ . This is also the case in Fig. 3, 5 and 12. 'CH4' specifies a substance, not a quantity. Use '$x_{CH4}$' or another |

appropriate quantity symbol. The same applies to Fig. 3 through 9 and Fig. 12.

**4  References**

[1]     IUPAC      Gold      Book,      term      'concentration',
        https://goldbook.iupac.org/terms/view/C01222

[2]     IUPAC      Gold      Book,      term      'mixing      ratio',
        https://goldbook.iupac.org/terms/view/M03948

[3]     IUPAC      Gold      Book,      term      'amount      fraction',
        https://goldbook.iupac.org/terms/view/A00296

[4]     GAW Report No. 242 19th WMO/IAEA Meeting on Carbon Dioxide, Other
        Greenhouse Gases and Related Measurement Techniques (GGMT-2017),
        https://library.wmo.int/doc_num.php?explnum_id=5456

[5]     SI    Brochure:    The    International    System    of    Units    (SI),
        https://www.bipm.org/en/publications/si-brochure/

[6]     WMO Guide to Meteorological Instruments and Methods of Observation,
        https://library.wmo.int/doc_num.php?explnum_id=4147

---

## Referee Comment (RC2) · Anonymous Referee #2 · 29 Jan 2020

This manuscript presents the results from a field deployment of a pair of low-cost metal-oxide sensors. The sensors were co-located with a reference instrument, allowing the researchers to train various calibration models to predict methane concentrations. These calibration models relied on the signals from the low-cost sensors as well as other sensors (i.e., temperature and humidity). Researchers then assessed the performance of and potential for these sensors using the predicted signals.

This manuscript is especially relevant to the field of low-cost sensor research and readers of Atmospheric Measurement Techniques for two reasons: (1) it provides an example of a long-term (multi-year) field deployment of low-cost metal-oxide sensors, and

(2) it provides an example of VOC sensors deployed to predict ambient methane levels - two areas that would benefit from further study. Furthermore, the deployment of the sensors in a remote area with little potential for the presence of confounding pollutants provides useful information on the potential ability of this sensor to be used for methane detection. Though a few revisions (listed below) are recommended prior to publication.

1. Please clarify throughout whether the results for the linear model being discussed in the text are based on the model that was fitted to the complete data set or the model which was fitted to the shorter training data set. Additionally, the training and testing periods defined for the linear model (in Table 1) and for the ANN (in the end of Section 2.4) appear to be different. Could the authors comment on the rationale for this choice and whether the use of these different periods might affect the comparability of the results for these two models presented in Table 1?

2. In Section 2.3, please provide information on any additional processing of the sensor data that may have occurred (e.g., filtering outliers, or removing sensor "warm-up" periods), or state that the data did not undergo additional filtering or processing.

3. Suggest moving the description of the motivation and development of the model for heat loss to an earlier point in the manuscript (e.g., after the description of the linear model in Section 3.0). This would assist the reader in their interpretation of the results in Table 1. Though the discussion of how this approach could be improved should remain in Section 3.5.

4. Could the authors provide additional information or discuss how the parameters of the model were selected (Eq. 2), for example, did this model yield substantial improvements over a simpler linear model?

5. Suggest expanding on the point made in Section 3.5 (Lines 253-254) to explain in what ways laboratory conditions over-simplify real-world conditions. This observation has been demonstrated in other studies [1, 2] and it could be valuable to highlight the challenges that may be associated with laboratory calibrations of sensors for this

particular application.

6. Could the authors provide additional detail on the potential or likelihood for confounding pollutants, in particular carbon monoxide (Section 2.2)? For example, are there any towns nearby where emissions from wintertime heating may be a concern, or did any major wildfires occur in the area throughout the deployment period?

7. Is there any concern that the temperature/humidity sensor described in Section 2.2 might itself experience any issues with drift or aging over such a long field deployment?

8. Line 38: add an 's', "assessment of low-cost sensor[s]"

9. Line 66: delete 'e.g.,', "in an area like e.g., the arctic"

10. Line 246-247: change the color of the red text to black

11. Line 254: delete 'it', "as it would be required"

References

[1] Castell, N., Dauge, F., Schneider, P., Vogt, M., Lerner, U., Fishbain, B., . . . Bartonova, A. (2017). Can commercial low-cost sensor platforms contribute to air quality monitoring and exposure estimates? Environment International, 99, 293-302.

[2] Piedrahita, R., Xiang, Y., Masson, N., Ortega, J., Collier, A., Jiang, Y., . . . Shang, L. (2014). The next generation of low-cost personal air quality sensors for quantitative exposure monitoring. Atmospheric Measurement Techniques Discussions, 7(2), 2425-2457.

---

## Author Comment (AC2) · 16 Feb 2020

**Response to Reviewer #2**

Reviewer feedback in copied in with black text, our response, how we plan to revise our manuscript, is given in indented blue text.

This manuscript presents the results from a field deployment of a pair of low-cost met-aloxide sensors. The sensors were co-located with a reference instrument, allowing the researchers to train various calibration models to predict methane concentrations. These calibration models relied on the signals from the low-cost sensors as well as

other sensors (i.e., temperature and humidity). Researchers then assessed the performance of and potential for these sensors using the predicted signals.

This manuscript is especially relevant to the field of low-cost sensor research and readers of Atmospheric Measurement Techniques for two reasons: (1) it provides an example of a long-term (multi-year) field deployment of low-cost metal-oxide sensors, and (2) it provides an example of VOC sensors deployed to predict ambient methane levels - two areas that would benefit from further study. Furthermore, the deployment of the sensors in a remote area with little potential for the presence of confounding pollutants provides useful information on the potential ability of this sensor to be used for methane detection. Though a few revisions (listed below) are recommended prior to publication.

> Thank you for your very supportive assessment. Your comments are very valuable for us to improve clarity and add an interesting aspect about the sensitivity of the TGS 2600 to CO from wildfires.

1. Please clarify throughout whether the results for the linear model being discussed in the text are based on the model that was fitted to the complete data set or the model which was fitted to the shorter training data set.

> This will be done.

Additionally, the training and testing periods defined for the linear model (in Table 1) and for the ANN (in the end of Section 2.4) appear to be different. Could the authors comment on the rationale for this choice and whether the use of these different periods might affect the comparability of the results for these two models presented in Table 1?

> We will recalculate the ANN to match the same data selection as we used for the linear model.

2. In Section 2.3, please provide information on any additional processing of the sensor data that may have occurred (e.g., filtering outliers, or removing sensor "warm-up" periods), or state that the data did not undergo additional filtering or processing.

    This will be done.

3. Suggest moving the description of the motivation and development of the model for heat loss to an earlier point in the manuscript (e.g., after the description of the linear model in Section 3.0). This would assist the reader in their interpretation of the results in Table 1. Though the discussion of how this approach could be improved should remain in Section 3.5.

    This can be done.

4. Could the authors provide additional information or discuss how the parameters of the model were selected (Eq. 2), for example, did this model yield substantial improvements over a simpler linear model?

    We used the stepAIC function of the MASS package in R. We will add the details to the text.

5. Suggest expanding on the point made in Section 3.5 (Lines 253-254) to explain in what ways laboratory conditions over-simplify real-world conditions. This observation has been demonstrated in other studies [1, 2] and it could be valuable to highlight the challenges that may be associated with laboratory calibrations of sensors for this particular application.

This aspect also caught the attention of Reviewer #1. We will thus revise the text, in part with advice from Dr. Nick Martin, National Physics Laboratories, London on this topic. In addition we will emphasize that testing in the temperature and moisture conditions in the range from –40 °C to 0 °C is not easy in a laboratory environment.

6. Could the authors provide additional detail on the potential or likelihood for confounding pollutants, in particular carbon monoxide (Section 2.2)? For example, are there any towns nearby where emissions from wintertime heating may be a concern, or did any major wildfires occur in the area throughout the deployment period?

We will add to the text that the study site is in such a remote site that winter time heating negligible, certainly not what one would expect from an urbanized area. However, wildfire influences in summer time (from fires in forests far to the south of our site and in the south side of the Brooks Range) may produce high CO levels that would lead to apparent high CH4 mole fractions. We will add an extra analysis of an episode where smoke from a wild fire south of the Brooks Range mountains was present at Toolik Field Station according to our own records, and compare that period with the period immediately before when the smoke arrived. We can also compare from similar weeks in the year before and year after when the fire occurred.

7. Is there any concern that the temperature/humidity sensor described in Section 2.2 might itself experience any issues with drift or aging over such a long field deployment?

Any sensor might be subject to drifting and aging. What we can do in our revisions is to compare our dedicated temperature and relative humidity sensor with the reference sensor of the long-term weather station at the same site.

[Figure]

8. Line 38: add an 's', "assessment of low-cost sensor[s]" 9. Line 66: delete 'e.g.,', "in an area like e.g., the arctic" 10. Line 246-247: change the color of the red text to black 11. Line 254: delete 'it', "as it would be required"

These minor changes will be applied as suggested.

**References**

[1] Castell, N., Dauge, F., Schneider, P., Vogt, M., Lerner, U., Fishbain, B., . . . Bartonova, A. (2017). Can commercial low-cost sensor platforms contribute to air quality monitoring and exposure estimates? Environment International, 99, 293-302.
[2] Piedrahita, R., Xiang, Y., Masson, N., Ortega, J., Collier, A., Jiang, Y., . . . Shang, L. (2014). The next generation of low-cost personal air quality sensors for quantitative exposure monitoring. Atmospheric Measurement Techniques Discussions, 7(2), 2425-2457.

---

## Author Response (AR1)

**Response to both reviewers**

In this feedback to reviewers we copied in the author response that we provided for each point mentioned by the reviewer and then added our response and how we revised the manuscript. Reviewer feedback is copied in with black text, our final author response and our plan to revise our manuscript (doi:10.5194/amt-2019-402-AC1 and doi:10.5194/amt-2019-402-AC2) is given in indented gray text, and our response of how we actually were able to do this is presented with blue text. Line numbers relating to the clean revised manuscript are given in blue font, followed by the line numbers in the track-changes version of the revised manuscript in red font.

**Response to Reviewer #1**

**1 General comments**

The manuscript summarises the findings from a 6-year field deployment of a small, low-cost methane sensor under low Arctic conditions. Given the interest of the community in small, low-cost sensors on the one hand and the measurement of atmospheric methane on the other hand, this work is of high relevance to the readers of Atmospheric Measurement Techniques.

The manuscript is written and structured well. The methods are described in appropriate detail. However, I see some shortcomings in the data analysis and presentation of results, detailed below, that should be addressed before publication.

> Thank you for your very valuable and detailed assessment, which definitely helps us improve our paper.

**2 Specific comments**

It is unclear which quantity the Authors use when they report the abundance of methane. The sentence 'We report all gas concentrations in mixing ratios by volume (ppm or ppb dry mole fractions)' (ll. 78–79) is contradictory, as it uses three distinct quantities as if they were synonymous. In a mixture of two components A and B, concentration c is defined as c = qA/(VA + VB) , where V is volume and q one of the quantities mass, amount, volume or number concentration [1]. Mixing ratio by volume r is rather uncommon and defined as r = VA/VB [2]. Mole fraction x (IUPAC recommends y for gaseous mixtures, but this is not common in atmospheric science) finally is defined as x = nA/(nA + nB), where n is amount of substance [3]. Given that the WMO scale for methane abundance is a mole fraction scale [4], the reporting of mole fractions would be desirable. While the use of the term 'concentration' for mole fraction is accepted for communication with the general public [4], a publication in a scientific journal should in my eyes favour exact terminology. In any case the Authors must make clear which quantity is reported.

> We apologize for the units confusion. We measured the dry mole fraction of methane (confirmed with our associate Dr. Martin Steinbacher from the Swiss Federal Labs for Materials Science and Technology, Empa), and we have replaced all other terms used to express methane units in the manuscript with "mole fraction" (defined in the first instance as dry mole fraction). We also now define ppm explicitly as $\mu$mol mol$^{-1}$ for clarity.

> Mole fraction is now used throughout the text, and the explicit clarification that the reference measurements are reported as dry mole fractions is given in Section 2.2 when specifying the LGR instruments (lines 77–80 [79–82]). Moreover, we now use $\mu$mol mol$^{-1}$ and nmol mol$^{-1}$ in text and figures.

Two TGS 2600 sensors were deployed at the site, referenced to as #1 and #2. However, in several instances in the manuscript 'TGS 2600' appears without a number when I think TGS 2600 #1 is meant. Also, 'sensor', 'TGS' and 'TGS2600' are used. This should be made more consistent. Results from TGS 2600 #2 are presented exclusively in l. 104, ll. 185–186 and Fig. 9. Explaining the minor role of TGS 2600 #2 around l. 59, l. 104 or l. 162 might prevent confusion of the reader.

> The paper is revised to be more explicit about the minor role of TGS 2600 #2. With only one sensor in place, an acceptable critique would have been that one sensor could be far off in its readings. Hence the second sensor #2 was only there for redundancy, and using this sensor actually showed that sensor #1 on which our analysis is primarily based, could be backed up by another (randomly selected) sensor of the same type.

We added the following text in Section 2.2 (lines 60–63 [61–64]):
*Sensor #1 is the primary sensor used in this study, whereas sensor #2 was only used as a replicate to simplify assessing potential problems with sensor #1. Because no such problems occurred, we will focus only on the results obtained with sensor #1 except for Section 2.2, where we used both sensors to assess their performance at weekly time resolution.*

ll. 74–77 How often and to which scale were the reference analysers calibrated?

Note that this is not a GAW reference station, but a seasonal research camp where we were able to check the LGR every summer season. We'll add this information and the numbers of the NOAA reference gas bottles we used. Before we had NOAA reference gases available it turned out that the accuracy of LGR was better than the specification of the available reference gases for calibration at the research station.

We added the following text to lines 92–96 [95–99]:
*Both FMA and the later FGGA analysers were used for eddy covariance applications, and thus the instruments were not calibrated as frequently as is done in applications for the Global Atmosphere Watch network (WMO, 2001). Both sensors were more accurate than the available calibration $CH_4$ gases at TFS. In 2015 for the first time it was possible to use a NOAA reference gas cylinder (#CB09827) to fine-tune the FGGA. This was typically done in the early summer season when field personnel arrived at TFS (late May).*

l. 153 '[...] relative humidity (which is a ratio and not a physical variable of atmospheric water content)' – I think I have an idea of what the authors mean, but I find the wording not quite right. Would the authors say that the refractive index of a material is not a physical variable because it is a ratio? In general, I miss some thoughts about the temperature dependence of the quantity used for expressing humidity.

We eliminated this statement about the ratio (line 172 [180–181]). A statement addressing the thoughts on temperature depencence of humidity was added in lines 301–304 [325–328]):
*Based on physical considerations one might expect that specific humidity or water vapor mixing ratio instead of absolute humidity could lead to further improvements, because absolute humidity still depends on temperature. However, our tests have not indicated a relevant gain of information or accuracy of prediction, but future work should also try to find a better physical correction model than the purely empirical one used here based on manufacturer information.*

The Figaro TGS 2600 has a heated sensing element, so the relative humidity at the sensing surface is different from the relative humidity in the environment. The temperature dependence of relative humidity makes this quantity a less than ideal choice for this type of correction. Unfortunately, both alternative quantities chosen by the authors to express water vapour content depend on temperature as well. Mixing ratio (by mass) or specific humidity would be temperature-independent alternatives [6]. Using the ideal gas equation, the terms in Eq. 2 that contain the product Ta·$\rho_v$ would also cancel out the temperature dependence of absolute humidity if Ta was absolute temperature (in K) – but in the manuscript a Celsius temperature is used. Hence, my suggestion to the authors is to try out either mixing ratio by mass or specific humidity as an independent variable in Eq. 2. Using absolute instead of Celsius temperature might be advisable as well.

We see the point, but one challenge is that such low-cost sensors are closer to a black-box system than to a fully controllable physical sensor. In addition, the manufacturer does not reveal all details of the sensor operation. In a conversation with Dr. Nick Martin, National Physics Laboratories, London, we learned that his lab has also explored the validity of using water concentration instead of relative humidity – in their laboratory tests they found an almost linear relationship between the two. We tried absolute humidity, vapor pressure, and relative humidity in our calculations, but absolute humidity always performed slightly better than the other two. We considered specific humidity and mixing ratio to be too close to absolute humidity and thus did not test this so far. Further tests could be run, but our approach is to adopt the same method as was used by previous authors (Collier-Oxandale et al., 2018; lines 258–260 in the discussion paper version).

We also understand the point that if we mix units of K or °C in the various equations (due to how the temperature dependence is expressed), we might introduce a constant bias (or at best just make the units messy). We can empirically test the question of whether temperature in K or °C gives the better results in our prediction models, and select the one that performs better for the revised version.

This is the parameter table from the linear model where temperature was entered in K instead of °C:

| | Estimate | Std. Error | t value | Pr(>\|t\|) |
|---|---|---|---|---|
| (Intercept) | 4.5259 | 0.1021 | 44.31 | <0.0001 |
| Rs_R0 | -0.4626 | 0.0247 | -18.73 | <0.0001 |
| I(1/Rs_R0) | -0.5538 | 0.1189 | -4.66 | <0.0001 |
| TA_AVG_M1B1_1.65_1_K | -0.0112 | 0.0004 | -26.86 | <0.0001 |
| RHO_V_CALC_M1B1_1.65_1 | -148.7784 | 5.7656 | -25.80 | <0.0001 |
| Rs_R0:TA_AVG_M1B1_1.65_1_K | 0.0019 | 0.0001 | 18.15 | <0.0001 |
| I(1/Rs_R0):TA_AVG_M1B1_1.65_1_K | 0.0037 | 0.0005 | 7.96 | <0.0001 |
| Rs_R0:RHO_V_CALC_M1B1_1.65_1 | 26.9994 | 4.9153 | 5.49 | <0.0001 |
| I(1/Rs_R0):RHO_V_CALC_M1B1_1.65_1 | -72.1729 | 15.4509 | -4.67 | <0.0001 |
| Rs_R0:TA_AVG_M1B1_1.65_1_K:RHO_V_CALC_M1B1_1.65_1 | 0.1939 | 0.0213 | 9.10 | <0.0001 |
| I(1/Rs_R0):TA_AVG_M1B1_1.65_1_K:RHO_V_CALC_M1B1_1.65_1 | 0.4949 | 0.0442 | 11.21 | <0.0001 |

And this is the model output where temperature was entered in K instead of °C **and** humidity was entered as mixing ratio instead of absolute humidity:

| | Estimate | Std. Error | t value | Pr(>\|t\|) |
|---|---|---|---|---|
| (Intercept) | 4.6552 | 0.1005 | 46.30 | <0.0001 |
| Rs_R0 | -0.4985 | 0.0239 | -20.83 | <0.0001 |
| I(1/Rs_R0) | -0.7269 | 0.1176 | -6.18 | <0.0001 |
| TA_AVG_M1B1_1.65_1_K | -0.0117 | 0.0004 | -28.59 | <0.0001 |
| MR_CALC_M1B1_1.65_1 | -113.5592 | 4.3851 | -25.90 | <0.0001 |
| Rs_R0:TA_AVG_M1B1_1.65_1_K | 0.0021 | 0.0001 | 20.24 | <0.0001 |
| I(1/Rs_R0):TA_AVG_M1B1_1.65_1_K | 0.0043 | 0.0005 | 9.46 | <0.0001 |
| Rs_R0:MR_CALC_M1B1_1.65_1 | 42.1642 | 3.8241 | 11.03 | <0.0001 |
| I(1/Rs_R0):MR_CALC_M1B1_1.65_1 | -71.0242 | 11.8599 | -5.99 | <0.0001 |
| Rs_R0:TA_AVG_M1B1_1.65_1_K:MR_CALC_M1B1_1.65_1 | 0.0709 | 0.0164 | 4.31 | <0.0001 |
| I(1/Rs_R0):TA_AVG_M1B1_1.65_1_K:MR_CALC_M1B1_1.65_1 | 0.4354 | 0.0339 | 12.84 | <0.0001 |

For comparison, the same output from the model that we use in Eq. (2) is copied in below:

| | Estimate | Std. Error | t value | Pr(>\|t\|) |
|---|---|---|---|---|
| (Intercept) | 1.4611 | 0.0172 | 84.97 | <0.0001 |
| Rs_R0 | 0.0597 | 0.0046 | 13.03 | <0.0001 |
| I(1/Rs_R0) | 0.4499 | 0.0162 | 27.82 | <0.0001 |
| TA_AVG_M1B1_1.65_1 | -0.0112 | 0.0004 | -26.86 | <0.0001 |
| RHO_V_CALC_M1B1_1.65_1 | -148.7784 | 5.7656 | -25.80 | <0.0001 |
| Rs_R0:TA_AVG_M1B1_1.65_1 | 0.0019 | 0.0001 | 18.15 | <0.0001 |
| I(1/Rs_R0):TA_AVG_M1B1_1.65_1 | 0.0037 | 0.0005 | 7.96 | <0.0001 |
| Rs_R0:RHO_V_CALC_M1B1_1.65_1 | 79.9518 | 1.8763 | 42.61 | <0.0001 |
| I(1/Rs_R0):RHO_V_CALC_M1B1_1.65_1 | 63.0215 | 4.5666 | 13.80 | <0.0001 |
| Rs_R0:TA_AVG_M1B1_1.65_1:RHO_V_CALC_M1B1_1.65_1 | 0.1939 | 0.0213 | 9.10 | <0.0001 |
| I(1/Rs_R0):TA_AVG_M1B1_1.65_1:RHO_V_CALC_M1B1_1.65_1 | 0.4949 | 0.0442 | 11.21 | <0.0001 |

As expected the model performance in all aspects did not change when the unit of the temperature measurements was changed from °C to K, but of course the parameter estimates changed. There was, however, no indication that the empirical interaction term in Eq. (2) cancels out because of the change in units. The last three lines in each table copied in above show these interaction terms. The parameters are $0.1939 \pm 0.0213$ and $0.4949 \pm 0.0442$ for the models using $\rho_v$, and $0.0709 \pm 0.0164$ and $0.4354 \pm 0.0339$ for the model using mixing ratio instead of $\rho_v$ (note that these parameter estimates do not change by changing the units used for temperature).

The overall model performances in a similar display as the one used in Table 1 in the manuscript looks like this:

| | Eq. (2) | | $T_a$ in K, not °C | | $MR$ not $\rho_v$; $T_a$ in K, not °C | |
|---|---|---|---|---|---|---|
| | Calibration | Validation | Calibration | Validation | Calibration | Validation |
| *Overall* | | | | | | |
| $R^2$ | 0.447 | 0.207 | 0.447 | 0.207 | 0.446 | 0.208 |
| RMSE ($\mu$mol mol$^{-1}$) | 0.026 | 0.041 | 0.026 | 0.041 | 0.026 | 0.041 |
| *Warm conditions ($T_a \geq 0°C$)* | | | | | | |
| $R^2$ | 0.518 | 0.288 | 0.518 | 0.288 | 0.519 | 0.289 |
| RMSE ($\mu$mol mol$^{-1}$) | 0.026 | 0.032 | 0.026 | 0.032 | 0.026 | 0.032 |
| *Cold conditions ($T_a < 0°C$)* | | | | | | |
| $R^2$ | 0.345 | 0.034 | 0.345 | 0.034 | 0.343 | 0.033 |
| RMSE ($\mu$mol mol$^{-1}$) | 0.027 | 0.052 | 0.027 | 0.052 | 0.027 | 0.052 |

Thus, using mixing ratio results in a marginally lower $R^2$ overall during the calibration period; although at temperatures above freezing mixing ratio is marginally performing better than $\rho_v$, the $R^2$ at temperatures below freezing is lower. Overall, this exercise confirms that the key physical processes

affecting the TGS 2600 are not well understood and hence we decided to keep using $\rho_v$. However, we added the following text to inform readers about this issue (lines 301–304 [325–328]):

*Based on physical considerations one might expect that specific humidity or water vapor mixing ratio instead of absolute humidity could lead to further improvements, because absolute humidity still depends on temperature. However, our tests have not indicated a relevant gain of information or accuracy of prediction, but future work should also try to find a better physical correction model than the purely empirical one used here based on (incomplete) manufacturer information.*

ll. 161–162 Using the entire dataset for estimating the parameters of Eq. 2 is a comprehensive test of how well the model can describe the dataset, but is of limited relevance for field deployments where calibrations are performed during limited periods of time and the main interest is in the uncertainty of independent measurements. For this reason, splitting the dataset into a calibration and a validation part yields important insights. The caption of Table 1 explains that the authors have in fact performed analyses of a split data set. This fact should also be mentioned in the main text around the lines given.

This point was also addressed by Reviewer 2. We will thus recalculate the ANN with the same selection for training vs. validation as we used for the linear model.

We did the reanalysis and this solved the issue with non-comparable presentations of goodness of fits due to different choices in calibration vs. validation periods. When done with the same selected periods the $R^2$ values of the ANN are lower during the calibration periods than our linear model (see updated Table 1), but are slightly higher during the validation period, except for warm conditions with temperatures above freezing.

l. 167 For the reasons given before, the results presented in the columns 'Linear Model - Calibration' and 'Linear Model - Validation' in Table 1 should be discussed here, even more so because the results for the validation period are substantially worse than for the period used for calibration.

This will be done.

This is where the ANN has a more even performance between calibration and validation period, although its performance in the calibration period was lower than was the performance of the linear model. Positively phrased one could say that the linear model makes more of the data that are available, but ANN learns more from the available data to make a prediction with validation data. Depending on what the aim is, both have benefits, and thus we added the following text (lines 205–209 [214–218]):
*When testing the linear model approach (Eq. 2) more rigorously by splitting the available data into a calibration period (years 2014–2016) and a validation period (years 2012–2013 and 2017–2018), some limitations can be seen, in particular under cold conditions where none of the approaches performed very well in the validation period. The ANN had a more balanced performance between calibration and validation period, although it performed slightly less well under warm conditions ($T_a \geq 0\,^\circ C$).*

ll. 193–194 Is there any conclusion that can be drawn from this finding of a -1:1 relationship?

The conclusion should be that both variability and stability can be improved at the same time because there is no tradeoff visible in Fig. 11. We will add text about this in the revised manuscript version.

We added the following text on lines 246–247 [255–256]:
*This indicates that both variability and stability can be improved at the same time because there is no tradeoff visible in Fig. 11.*

Sect. 3.4 The discussion in Sect. 3.1 leans heavily on the coefficient of determination R2. In Table 1, each R2 for the ANN approach is higher than the corresponding R2 for the Linear Model. Considering just the validation period, the ANN approach outperforms the Linear Model by a factor of 3–10 by this measure. Similarly, in Fig. 6 and 8 the ANN approach outperforms the Linear Model (comparing R2 of 'ANN' and 'c/v', 'all' is irrelevant in this respect); in Fig. 5 and 7 they perform nearly equally well. None of these comparisons is made here. Instead, the authors state that the root mean square error (RMSE) does not improve substantially with the ANN approach. While I generally appreciate the reporting of RMSE together with R2, its interpretation here is questionable.

The reviewer is right, $R^2$ in an ANN is not the same as $R^2$ in a linear model and thus comparing the numbers may be misleading. We will revise this and provide a goodness-of-fit statistic that is comparable between a linear model and ANN. RMSE is of course the better comparison because it has the same meaning in cases where either linear or nonlinear fits are used and evaluated.

We homogenized the computation of the $R^2$ values for both methods, which however was not the primary issue here; the biggest change was related to the different selection of calibration vs. validation periods. How the values given in Table 1 are directly comparable and relate to the same selection of calibration vs. validation period. The text on lines 159–160 [166–168] was updated accordingly.

On the one hand, RMSE is reported in Table 1 with one significant digit only, potentially masking up to $\sim$30% differences for an RMSE of 0.03 ppm (0.025 ppm vs. 0.0349 ppm). On the other hand, the RMSE should be seen in the context of the variability of the data, specifically the root mean square difference between the reference measurements and their mean value over the whole dataset, which is not stated. Overall, the discussion in this section appears negatively biased with regard to the ANN approach. This also manifests in the last paragraph of this section, where 'understand[ing] the physical response of TGS sensors' is prioritised over 'technically nicer fits to data', a stark contrast to the lack of a physical interpretation of the terms in the empirical model (Eq. 2). Section 3.4 must be revised to reach the level of neutrality expected from a scientific publication.

Thank you for these suggestions; we will remove the wording used in the conclusion about neural networks, and rethink how we can best compare the two models both with unbiased estimators of their fit to the data and with their usefulness in interpreting the data in mind. We will also adjust the significant digits of the measures we use (e.g., RMSE) to make them more comparable and useful.

We now show one additional digit in both $R^2$ and RMSE values. We also added the extra digit in the $R^2$ lines to allow the reviewer to compare the minor differences between some values that would otherwise be hidden in the rounding to the second digit. In Figures 5–8 we added RMSE below $R^2$.

Section 3.4 was revised to be more neutral and reflect the conditions that when the same calibration and validation periods are used with both approaches, then the apparent outperformance of ANN is less obvious. We agree that for many other applications there is a high potential of using an ANN but not really in this context where we try to separate the influence of $T_a$ and $\rho_v$ on the sensor signal, while at the same time of course $T_a$ and $\rho_v$ influence the production of methane in anaerobic soils and thus modify the $CH_4$ mole fractions to be measured in ambient air.

We thus rewrote Section 3.4 accordingly (see lines 274–289 [284–312]).

Please take note that in the previous submission we had used ca. five years of calibration period vs. only 1.5 year of validation for the ANN, whereas we used the same setting as in the revised version for the linear model (3 calibration years vs 4 validation years). Please be aware that while we have noticed good performance of ANNs in other applications, we do have some reservations for applications as the one presented here.

ll. 229–231 To make such an argument, the reader must be informed about the amplitude of all input variables, especially SC.

In light of the comments above we will reword or delete this statement.

We deleted this statement since with reduced coverage of winter conditions by the calibration period (to make the linear model and ANN approaches comparable) the ANN performance deteriorated substantially.

l. 237 I might be mistaken, but as far as I understand the term homoscedasticity it would in this case mean that the variance of the deviation in CH4 abundance is the same for every temperature bin. The authors do not report variances, but both interquartile range and 95% confidence interval suggest that the variance is higher at low temperatures than at high temperatures, i.e. heteroscedasticity.

What we wanted to say is that the variances (shown with the 95% CI) are quite constant over a wide range of temperatures and absolute humidity. We of course agree that if one focuses at the difference between one bin and another there are many comparisons where the statement is not 100% correct, even when the Bonferroni correction for multiple testing is considered. In comparisons that span a difference of more than 10–15 °C, the within-bin variances are of course not always homoscedastic. We will therefore reword our description of this point to avoid this confusion.

The new wording (lines 291–293 [314–317]) is:
*The inter-quartile ranges and 95% confidence intervals of each air temperature (Fig. 12a) or absolute humidity bin (Fig. 12b) are very similar over a wide range of temperature and humidity, but tend to become more variable in bins with few data (i.e., lowest and highest temperatures, and highest absolute humidities in Fig. 12).*

ll. 253–257 'laboratory conditions simplify the real world too much' – What could be the simplification that makes laboratory calibrations problematic? The input variables used in the empirical model (Eq. 2) can – practical difficulties taken aside – be controlled in the lab. Any other variable that might prevent transfer of lab results to field conditions is not included in the empirical model, so the problem would not be a simplification of the lab environment but a model deficiency. The last sentence of the paragraph seems to go in this direction ('relevant factors'), but is unclear. Please explain better or leave out.

This statement also caught the attention of Reviewer #2, and given both reviewer comments we will revise the text. As the reviewer says, there are several ways in which laboratory and real world conditions apply to sensor calibrations. For example, in our revision we will emphasize that testing in the temperature and moisture conditions that exist in the range from –40 °C to 0 °C is not easy in a laboratory environment. In addition, we have discussed this general point with Dr. Nick Martin, National Physics Laboratories, London, and will adopt some of his perspective and advice in our revised text.

We revised the text to read (lines 313–320 [338–351]):
*While this is theoretically correct, it remains difficult to carry out laboratory treatments from –41 °C to 27 °C as would be required for our Arctic site. The data we present indicate that most likely it is absolute humidity (or specific humidity or mixing ratio), not relative humidity, that should be used for such calibrations, which in principle should provide the best quality results if the relevant factors are known and can be included in the calibration set-up. It would be desirable that manufacturers carry out both laboratory tests and field trials and provide the necessary correction functions together with sensors. However, due to the expense and time it takes to carry out long tests, be it in the laboratory or in the field, the present development goes in the direction of collocation studies en route to certification of sensors (N. Martin, NPL, UK, pers. comm.), similar to what we have done in the Arctic.*

ll. 258–261 Suggesting to move the first sentence to l. 150 and to remove the other one (repetition).

This will be done.

The second sentence was deleted [344–348], and the first was moved to lines 168–170 [177–178].

Fig. 5–8 The graphs are squeezed in horizontal direction, making comparisons between the lines difficult. A shorter period, e.g. 14 days, would give more insight.

We agree that the graphs are squeezed, but we are concerned that expanding them horizontally would invite comparisons over shorter time periods that are not the focus of the study. For example, on lines 179–181 we wrote "The TGS 2600 is not expected to provide short-term accuracy comparable to high-quality instrumentation (see also Lewis et al., 2018). However, Eugster and Kling (2012) argued that such measurements still may provide additional insights as compared to the passive samplers described by Godbout et al. (2006a, b), integrating over longer time frames". Given that, we tried to draw the attention of the reader to the fact that such a low-cost sensor is not expected to provide high accuracy over the short-term. Zooming in would thus highlight the opposite of what we consider meaningful in this study. However, we will publish the original data along with our paper (we will however select a different repository than the original mentioned under "Data availability"), which will allow the interested reader to download the data and zoom in as desired.

The data set will be made available via DOI:10.6073/pasta/dddeb05b2806e2f5788fadd6fc590ef1, and the fits shown in Figures 5–8 will be made available via DOI:10.3929/ethz-b-000369689. The text under "Data availability" was updated accordingly. Note that the second DOI will only be published after acceptance of the manuscript.

Fig. 5 and 7 The collected in 2012 and 2015 are both part of the calibration period, not the validation period, which is important to know for the reader to correctly interpret 'TGS 2600 c/v' and 'ANN'. I therefore strongly suggest a note in the figure caption.

This information will be added.

We added the information *This example belongs to the validation period of the TGS 2600 c/v and ANN fits* to the captions of Fig. 5–8. Note that only the ANN in the previous version was using 2012 and 2015 for calibration. In the revised version both approaches show validation period results in all four figures. Thanks for pointing this out.

Fig. 5 Suggesting to replace '(TGS 2600 - Reference)' with '(TGS 2600 all - Reference)' in the caption

This will be done.

Done.

Fig. 9 A plot of the difference of the methane abundance calculated from the measurements of the two sensors would be of high interest for the readers. With such a new panel it is also important to state if the parameters derived for TGS 2600 #1 have been used when applying Eq. 2 to the measurements of TGS 2600 #2. In my opinion the new panel could replace panel (b), as the signal difference seems of little relevance.

We will replace panel (b) that currently shows the relative differences with a new panel showing the absolute differences as suggested by the reviewer. And we will more clearly state that we used the parameters derived for sensor #1 also for sensor #2 (as correctly interpreted by this reviewer).

We replaced panel (b) but then rearranged (b) and (c), so that the absolute difference between the two sensors is now shown in panel (c). And we added the information *The signals from both sensors were converted to CH$_4$ using Eq. (2) parameterized with data from TGS sensor #1* to the caption.

Fig. 10 If the main text in ll. 189–190 is correct, 'and the reference' is missing at the end of the first sentence of the figure caption.

Yes, this will be added.

Done.

**3 Technical corrections**

l. 1 Suggesting to remove "weak" to avoid misunderstanding. Alternatively, it could be written in parentheses like it the conclusions.
Done.

l. 8 Insert a space between value and unit of temperature. This correction is necessary wherever '°C' is used [5].
Done.

l. 76 replace**d**
Done.

l. 140 typeset 'Ta' as Ta
We assume this relates to former line 149, there we found an occurrence of Ta that should be $T_a$ which we corrected.

l. 305 'cross-sensitivities [...] **are** of no concern'
Done.

Fig. 2 There seems to be a non-displayable glyph at the beginning of the label for the vertical axis, possibly a $\Delta$. This is also the case in Fig. 3, 5 and 12.
We could not reproduce this problem in Acrobat or Preview on Mac. Could it be an issue with the PDF reader you're using? If you can specify with which reader you use to see this problem, then we will double-check at print stage to make sure the PDF has all fonts correctly embedded. Please also specify on which operating system you see this problem (Windows, Linux, iOS, Android, MacOS, . . . ).

'CH4' specifies a substance, not a quantity. Use '$x_{CH4}$' or another appropriate quantity symbol. The same applies to Fig. 3 through 9 and Fig. 12.
We now use $\chi_{CH_4}$ in all figures as suggested by this reviewer. Please use Acrobat Reader for double-checking, because if $\Delta$ is not correctly seen in your tool, then most likely the same might be the problem with $\chi$. But we'll solve this issue at print stage if we can reproduce the problem. Normally this is a mismatch of PDF file type versions. Our figures use PDF version 1.4, but this could be changed if needed (although AMT also uses PDF version 1.4 for their file versions, which is archivable PDF of type PDF/A-1 and thus should be correctly displayed in all readers, hence our question for the details to reproduce the problem observed by this reviewer).

These will be incorporated in the revised manuscript, thank you for highlighting these items.

**Response to Reviewer #2**

This manuscript presents the results from a field deployment of a pair of low-cost metaloxide sensors. The sensors were co-located with a reference instrument, allowing the researchers to train various calibration models to predict methane concentrations. These calibration models relied on the signals from the low-cost sensors as well as other sensors (i.e., temperature and humidity). Researchers then assessed the performance of and potential for these sensors using the predicted signals.

This manuscript is especially relevant to the field of low-cost sensor research and readers of Atmospheric Measurement Techniques for two reasons: (1) it provides an example of a long-term (multi-year) field deployment of low-cost metal-oxide sensors, and (2) it provides an example of VOC sensors deployed to predict ambient methane levels - two areas that would benefit from further study. Furthermore, the deployment of the sensors in a remote area with little potential for the presence of confounding pollutants provides useful information on the potential ability of this sensor to be used for methane detection. Though a few revisions (listed below) are recommended prior to publication.

> Thank you for your very supportive assessment. Your comments are very valuable for us to improve clarity and add an interesting aspect about the sensitivity of the TGS 2600 to CO from wildfires.

1. Please clarify throughout whether the results for the linear model being discussed in the text are based on the model that was fitted to the complete data set or the model which was fitted to the shorter training data set.

> This will be done.

> We added the following text to clarify (lines 186–189 [195–198]):
> *Unless explicitly mentioned, we analysed $CH_4$ mole fractions computed with Eq. (2) using the parameters obtained from all data measured by TGS 2600 sensor #1. Only in the direct comparison with the ANN (Section 3.1) did we determine an additional parameter set using the same calibration period as the ANN used, so that a direct comparison of performance in validation was possible.*

Additionally, the training and testing periods defined for the linear model (in Table 1) and for the ANN (in the end of Section 2.4) appear to be different. Could the authors comment on the rationale for this choice and whether the use of these different periods might affect the comparability of the results for these two models presented in Table 1?

> We will recalculate the ANN to match the same data selection as we used for the linear model.

> Done. See also our response to reviewer #1 on this issue.

2. In Section 2.3, please provide information on any additional processing of the sensor data that may have occurred (e.g., filtering outliers, or removing sensor "warm-up" periods), or state that the data did not undergo additional filtering or processing.

> This will be done.

> We added the following text (lines 103–110 [106–113]):
> *Before analyses the data were processed in the following way: (1) outliers were removed; (2) relative humidities $> 105\%$ (accuracy of capacitive humidity sensors) were deleted; (3) reference $CH_4$ mole fractions obtained from the FGGA (since 2016) were filtered based on hard boundaries of housekeeping variables available for quality control. For the latter we used the following hard boundaries for filtering: (a) sample cell pressure had to be in the range 130–143 Torr; (b) the instrument-specific*

*ringdown time of the laser for CH$_4$ measurements had to be in the range 13–17 ms. The accepted reference CH$_4$ mole fractions were thus all measured in the narrow range of cell pressures between 139.7 and 140.3 Torr and laser ringdown times between 14.02 and 14.94 ms, which indicates best performance of the analyser. Before 2016 (FMA instrument) these house-keeping variables were not recorded.*

3. Suggest moving the description of the motivation and development of the model for heat loss to an earlier point in the manuscript (e.g., after the description of the linear model in Section 3.0). This would assist the reader in their interpretation of the results in Table 1. Though the discussion of how this approach could be improved should remain in Section 3.5.

This can be done.

We moved former lines 268–278 to Section 3 (now lines 192–202 [201–211]) with an additional introductory sentence: *If ambient temperature influences the signal of the TGS 2600 in such a way as expected from the technical documentation (Figaro, 2005b,a), then wind speed could be a third factor influencing the conversion from TGS 2600 sensor voltages to CH$_4$ mole fractions.*

In the gap from where we moved the text away we inserted (lines 327–328 [358–368]): *To address this additional factor, we used the heat loss model given in Eq. (3).*

4. Could the authors provide additional information or discuss how the parameters of the model were selected (Eq. 2), for example, did this model yield substantial improvements over a simpler linear model?

We used the stepAIC function of the MASS package in R. We will add the details to the text.

We added the text (lines 184–186 [193–195]):
*The linear model in Eq. (2) was derived from a suite of candidate models including interactions among predictors and including quadratic terms of each variable, and then stepwise elimination using the stepAIC function in the MASS package of R was employed to find the model with the lowest AIC (Akaike's Information Criterion).*

5. Suggest expanding on the point made in Section 3.5 (Lines 253-254) to explain in what ways laboratory conditions over-simplify real-world conditions. This observation has been demonstrated in other studies [1, 2] and it could be valuable to highlight the challenges that may be associated with laboratory calibrations of sensors for this particular application.

This aspect also caught the attention of Reviewer #1. We will thus revise the text, in part with advice from Dr. Nick Martin, National Physics Laboratories, London on this topic. In addition we will emphasize that testing in the temperature and moisture conditions in the range from –40 °C to 0 °C is not easy in a laboratory environment.

Please see our response to the same point of Reviewer #1.

6. Could the authors provide additional detail on the potential or likelihood for confounding pollutants, in particular carbon monoxide (Section 2.2)? For example, are there any towns nearby where emissions from wintertime heating may be a concern, or did any major wildfires occur in the area throughout the deployment period?

We will add to the text that the study site is in such a remote site that winter time heating negligible, certainly not what one would expect from an urbanized area. However, wildfire influences in summer time (from fires in forests far to the south of our site and on the south side of the Brooks Range) may produce high CO levels that would lead to apparent high CH4 mole fractions. We will add an extra analysis of an episode where smoke from a wild fire south of the Brooks Range mountains was present at Toolik Field Station according to our own records, and compare that period with the period immediately before the smoke arrived. We can also compare from similar weeks in the year before and year after when the fire occurred.

We have written records from the smell of smoke from wildfires south of the Brooks Range that affected the measurements on 26 June 2015. Thus, we carried out a case study analysis for that day and show the day of interest (white background) with the day before and the day after (gray background):

[Figure]

As an insert we plotted the histograms of the difference between the two TGS-derived CH₄ mole fractions and the ANN fit. Then, we carried out the same analyses for four other cases: (1) three days before the wildfire event, and (2–4) the same date one, two and three years later:

[Figure]

The general impression is that the TGS-derived $CH_4$ mole fractions are roughly $0.03\,\mu\text{mol}\,\text{mol}^{-1}$ reduced by the increased carbon monoxide mole fractions and other air pollutions during the second part of the day. The mean differences $\pm$ standard deviations are given in the left part of each panel. This offset does not differ between TGS #1 and TGS #2, but it should be considered a coincidence that using Eq. (2) that was parameterized with data from TGS #1 performs better with TGS #2 in all comparison case studies. The variability of the difference also appears to be slightly larger ($\pm0.025$ to $\pm0.027$ on 26 June 2015 vs. $\pm0.014$ in all four comparisons).

Thus, in the text we added the following information (lines 221–228 [230–237]):
*Because of the absence of local sources of carbon monoxide and other air pollutants to which the TGS 2600 sensor is also sensitive besides $CH_4$, we investigated a special case when smoke and haze from wildfires south of the Books Range polluted the air in the TFS area on 26 June 2015 and compared the performance of both TGS sensors during that day with conditions three days before that event, and on the same date in the following three years. The net effect of increased air pollutants was an apparent small decrease of the $CH_4$ mole fractions calculated via Eq. (2) by approximately − $0.03\,\mu\text{mol}\,\text{mol}^{-1}$. At the same time the variability of the residuals increased from typically $\pm0.014$*

*to $\pm0.027\,\mu mol\,mol^{-1}$ (24-hour averages). Thus, the influence of the wildfire smoke was of the same order of magnitude as the difference between TGS-derived $CH_4$ mole fractions and the reference instrument on most other days of the year (see Figs. 5–8).*

7. Is there any concern that the temperature/humidity sensor described in Section 2.2 might itself experience any issues with drift or aging over such a long field deployment?

Any sensor might be subject to drifting and aging. What we can do in our revisions is to compare our dedicated temperature and relative humidity sensor with the reference sensor of the long-term weather station at the same site.

The HMP45AC sensor that we used for the long-term temperature and relative humidity measurements has a sensor head that can be exchanged against a calibrated one to avoid long-term drifts in the measurements. We exchanged the head every ca. 3 years to minimise long-term drift errors of our temperature and relative humidity measurements. We added this information to the corresponding text (lines 87–89 [91–92]:
*The factory-calibrated HMP45AC sensor head was exchanged against a newly calibrated one every ca. 3 years to minimize long-term drift effects in temperature and relative humidity measurements (J. Laundre, pers. comm.).*

We initially thought that a comparison with the temperature and relative humidity sensors of the long-term weather station at TFS (roughly 0.5 km to the north-east) would be the way to go, and hence our Final Author Response (in gray above). But when comparing the data we realized that the microclimatic difference between TFS close to Toolik Lake and our inland site at some distance from the lake shore and at higher elevation in the wet tundra show important seasonal trends that rather look like real differences, and thus do not really allow to test for drifts. Thus, having replaced the calibrated sensor head every ca. 3 years is probably the best we could do to minimize concerns about sensor drift.

8. Line 38: add an 's', "assessment of low-cost sensor[s]" 9. Line 66: delete 'e.g.,', "in an area like e.g., the arctic" 10. Line 246-247: change the color of the red text to black 11. Line 254: delete 'it', "as it would be required"

These minor changes will be applied as suggested.

Done.

**References**

[revised manuscript text omitted]